# Dorsal periaqueductal gray ensembles represent approach and avoidance states

**Fernando MCV Reis[1‡]\*, Johannes Y Lee[2‡], Sandra Maesta-Pereira[1], Peter J Schuette[1], Meghmik Chakerian[1], Jinhan Liu[2], Mimi Q La-Vu[1], Brooke C Tobias[1], Juliane M Ikebara[3], Alexandre Hiroaki Kihara[3], Newton S Canteras[4], Jonathan C Kao[2†]\*, Avishek Adhikari[1†]\***

[1]Department of Psychology, University of California, Los Angeles, Los Angeles, United States; [2]Department of Electrical and Computer Engineering, University of California, Los Angeles, Los Angeles, United States; [3]Centro de Matemática, Computação e Cognição, Universidade Federal do ABC, São Bernardo do Campo, São Paulo, Brazil; [4]Department of Anatomy, Institute of Biomedical Sciences, University of São Paulo, São Paulo, Brazil

**Abstract** Animals must balance needs to approach threats for risk assessment and to avoid danger. The dorsal periaqueductal gray (dPAG) controls defensive behaviors, but it is unknown how it represents states associated with threat approach and avoidance. We identified a dPAG threatavoidance ensemble in mice that showed higher activity farther from threats such as the open arms of the elevated plus maze and a predator. These cells were also more active during threat avoidance behaviors such as escape and freezing, even though these behaviors have antagonistic motor output. Conversely, the threat approach ensemble was more active during risk assessment behaviors and near threats. Furthermore, unsupervised methods showed that avoidance/approach states were encoded with shared activity patterns across threats. Lastly, the relative number of cells in each ensemble predicted threat avoidance across mice. Thus, dPAG ensembles dynamically encode threat approach and avoidance states, providing a flexible mechanism to balance risk assessment and danger avoidance.

**\*For correspondence:**
fmcvreis.84@gmail.com (FMCVR);
kao@seas.ucla.edu (JCK);
avi@psych.ucla.edu (AA)

[†]These authors also contributed equally to this work
[‡]These authors also contributed equally to this work

**Competing interests:** The authors declare that no competing interests exist.

## Introduction

Behavioral variables and emotional states are thought to be represented in neural activity (for review, see *Salzman and Fusi, 2010*; *Anderson and Adolphs, 2014*). Such representations must be specific enough to differentiate across behaviors, yet general enough to maintain functional cohesion across diverse threatening situations (*Gründemann et al., 2019*). A large body of evidence has shown that defensive behaviors related to threat exposure are represented in dorsal periaqueductal gray (dPAG) activity (*Deng et al., 2016*; *Esteban Masferrer et al., 2020*; *Evans et al., 2018*; *Watson et al., 2016*), as dPAG activity correlates with escape and freeze. Additionally, dPAG optogenetic and electrical stimulation induce these behaviors, as well as aversion (*Brandão et al., 1982*; *Carvalho et al., 2015*; *Carvalho et al., 2018*; *Deng et al., 2016*; *Tovote et al., 2016*). Furthermore, pharmacological manipulations of dPAG activity impact open arm exploration in the elevated plus maze (EPM), a traditional measure of rodent anxiety (*Fogaça et al., 2012*). Lastly, PAG activity in humans correlates positively with threat imminence (*Mobbs et al., 2007*; *Mobbs et al., 2010*). These reports show that the dPAG is a central node orchestrating defensive responses.

However, it is unknown how the dPAG represents moment-to-moment changes in brain states during threat exposure. The two main behavioral states observed during exposure to threats are approach and avoidance (*Stankowich, 2019*). In the approach state, animals voluntarily go near the threat and perform risk assessment behaviors. In this state, the exploratory risk evaluation drive is

stronger than the motivation to avoid danger. By contrast, in the avoidance state, animals perceive high risk, and thus attempt to minimize exposure to danger by escaping, freezing, and maintaining distance to the threat. No reports to date have investigated whether the dPAG consistently encodes approach and avoidance states across distinct threats.

Key questions regarding the neural representation of approach and avoidance states remain unanswered. Do dPAG cells respond uniformly to transitions between higher and lower threat imminence? What is the overlap between the dPAG encoding of two completely distinct threats? Does dPAG neuronal activity encode moment-to-moment changes regarding defensive approach and avoidance states? Addressing these questions would require population-level analysis of dPAG cells recorded longitudinally across threat modalities. Here, we report experimental data and analyses that directly address these questions.

## Results and discussion

We performed microendoscopic calcium imaging of dPAG neurons expressing GCaMP6s (*Figure 1A* and *Figure 1—figure supplement 1*; *Cai et al., 2016*) during EPM and rat exposure assays. During the EPM test, we recorded 107 ± 19 cells per mouse (*n* = 8 mice; 857 cells were imaged; see Materials and methods). As expected, mice spent more time in the closed arms of the EPM (*Figure 1B*). They also displayed exploratory head dips (average frequency of events = 27.0 ± 9.7; *n* = 8 mice) over the edges of the open arms (*Figure 1B–C*). During EPM exploration cells often showed preferential activity either in closed or open arms (*Figure 1D–E*). To identify EPM arm-type modulated neurons, we defined an 'arm score metric' ranging from −1 to +1, in which the +1 indicates that cell activity in the open arm is greater than activity in the closed arm, and vice versa. The arm score distribution in the observed data is wider than expected by chance, indicating that dPAG cells show robust preference for EPM arm types (*Figure 1F*, left panel). We defined neurons as belonging to one of the ensembles if activity in each arm-type was significantly greater than the pooled activity in the opposite arm type (*Figure 1F*, right panel, see Materials and methods). The results showed that cells fired similarly in arms of the same type, as cell activity in one open arm was highly correlated to activity in the other open arm (*Figure 1G*, top panel). Conversely, cell activity in closed and open arms was negatively correlated (*Figure 1G*, bottom panel).

Based on the distribution of cells per arm score, roughly half of the dPAG neurons were classified as arm-modulated cells (49%, with 26% closed- and 23% open-modulated cells) (*Figure 1H*), which suggests these ensembles are functionally relevant dPAG populations. During transitions between arms, we identified opposite changes in activity levels of these two major, non-overlapping populations of dPAG neurons (*Figure 1I* and *Figure 2A–B*). To ensure that positive results were not due to cell categorization itself, in *Figure 2A–C,G*, cell categorization was done on training data and results were plotted for separate testing data (see Materials and methods). For example, the closed arm-activated ensemble showed a decrease in activity when mice traversed from a closed arm to an open arm. Moreover, open and closed cells showed increased and decreased activity, respectively, during exploratory head dip behavior (*Figure 2C*). Aggregate activity of dPAG cells in the EPM did not show significantly higher activity in the open arms, showing that in the entire population, the closed and open cell patterns counterbalance each other (*Figure 2—figure supplement 1C*). Importantly, 95% of dPAG cells showed relatively low correlations of speed and neural activity, between −0.17 and +0.2. These results suggest that arm-related activity preferences are not driven by variations in velocity (*Figure 2D*). Instead, if EPM arm type is prominently represented in dPAG ensemble activity, then it may be possible to use dPAG activation patterns to differentiate mouse location in the EPM. Indeed, upon training support vector machine (SVM) decoders on dPAG activity, we obtained significantly higher than chance performance in identifying whether the mouse was in an open or closed arm (*Figure 2E*, see Materials and methods).

Next, we investigated whether dPAG activity could also predict specific mouse positions within the arms. An acceptable interpretation of EPM behavior is that mice avoid the open arms because they are more vulnerable to potential threats in open spaces (*Montgomery, 1955*; *Walf and Frye, 2007*). This view suggests that the beginning of the open arms is only slightly threatening, as if a dangerous stimulus is detected the mouse can quickly retreat to the safety of the closed arms. Conversely, it takes a longer time for the mouse to retreat to the closed arms if they are at the extreme end of the open arms. Thus, we argue that distance from the center of the maze is related to a

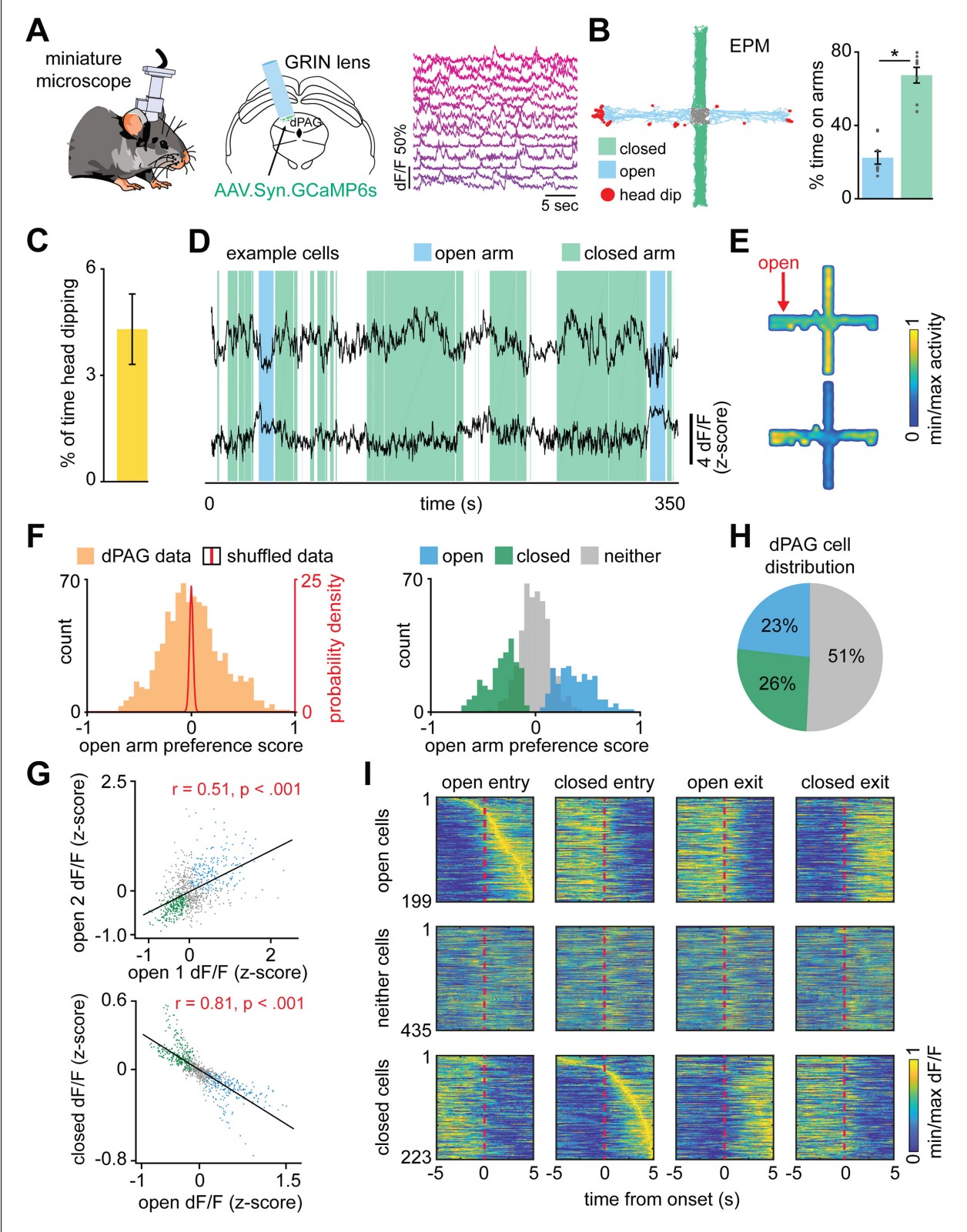

**Figure 1.** Dorsal periaqueductal gray (dPAG) neuronal ensembles encode arm type in the elevated plus maze (EPM). (**A**) Gradient refractive index (GRIN) lens implantation, virus expression strategy, and example Ca$^{2+}$ signals of neurons in the dorsal periaqueductal gray (dPAG). (**B**) Example mouse exploration path recorded in the EPM. Mice spent significantly more time in the closed arms compared to the open arms (data are represented as mean ± SEM; $W = 0$, p=0.012, Wilcoxon signed rank test, $n = 8$ mice). (**C**) Mean percentage of time in which mice engaged in head dips during the

*Figure 1 continued on next page*

Figure 1 continued

whole EPM session (n = 8 mice). (D) dPAG dF/F traces from the same mouse that display preferential activity in the closed (upper trace) and open (lower trace) arms of the EPM (open and closed arm-preferring cells). Epochs corresponding to exploration of the closed and open arms are shown respectively as green and blue shaded areas. (E) Activity heatmaps for corresponding example neurons shown in (D). (F) The open arm preference score was calculated for each neuron (orange bars; see Materials and methods), as was the distribution of open arm preference score for shuffled data (red line). Bars show the distribution of open arm preference score for open, closed, and neither cells (n = 857 cells). (G) Scatterplots showing correlations between neural activity across the two open arms (top) and between open and closed arms of the EPM (bottom). Each point represents one cell (n = 857 cells, r = Pearson's correlation coefficient). (H) Pie chart shows the percent of all recorded neurons that were classified as open, closed, or neither cells (n = 857 cells). (I) For each subplot, each row depicts the mean normalized activity of an open, closed, or neither arm-preferring cell during behavior-aligned arm transitions (n = 857 cells).

The online version of this article includes the following figure supplement(s) for figure 1:

**Figure supplement 1.** Deep brain imaging of dorsal periaqueductal gray (dPAG) neurons and distribution of elevated plus maze (EPM) scores.

gradient of threat, with the highest threat levels at the end of the open arms. The animal's behavior supports this view, as mice avoid the extreme end of the open arm more robustly than the beginning of the open arm (*Figure 2—figure supplement 1B*). We therefore defined the EPM location index to be a linearly varying metric between −1 and +1 (at closed and open arms extremities, respectively), assigning 0 to the center of the maze (*Figure 2F*, see Materials and methods). We then fitted a linear regression model to predict the EPM location index from dPAG cell activity. Interestingly, the model showed significantly higher than chance performance in predicting EPM location index, suggesting that the identified dPAG ensembles may not only encode arm type, but rather a risk perception and threat exposure gradient (*Figure 2F–G* and *Figure 2—figure supplement 1A*). Though there are closed and open arm-preferring cells, aggregate bulk dPAG activity did not show different activation levels across EPM arms (*Figure 2—figure supplement 1C*).

The above results show that the identified ensembles encode approach and avoidance toward the threatening locations in the EPM. To investigate whether dPAG population coding of approach-avoidance states generalizes to exploratory behavior across different threatening contexts, we recorded the same dPAG neurons during exposure to a live predator (*Figure 3A–B* and *Figure 3—figure supplement 1*). Mice were allowed to freely explore a context, a long chamber, in the presence of a rat, which was tethered with a harness to one end of the chamber (see Materials and methods). All behavioral data from synchronized videos underwent automated behavior scoring (*Mathis et al., 2018*). Mice spent most of the trial away from the rat (*Figure 3A*), indicating aversion from perceived threat. Consistent with previous threat imminence theories and the array of defensive behaviors evoked in the presence of a predator (*Blanchard et al., 2011*; *McNaughton and Corr, 2004*; *Perusini and Fanselow, 2015*; *Stankowich, 2019*), mice presented defensive strategy repertoires composed of approach and avoidance-related behaviors (i.e., escape and freeze) (*Figure 3A*). Freeze bouts lasted 2.1 ± 1.3 s on average. Notably, average dPAG activity increased with rat proximity, rat movement onset, and escape, and decreased during approach, while no significant change was observed during freezing (*Figure 3C–E*). Importantly, mice displayed no signs of aversion nor differences in dPAG activity with proximity to a toy rat (*Figure 3—figure supplement 2*).

We then explored whether the activity of dPAG closed and open cell ensembles identified in the EPM also represent risk evaluation in the rat assay. A positive result would show that the ensembles are likely responding not only to the original sensory biases (i.e., closed and open arms features), but potentially representing behavioral states that generalize across threats. Indeed, open cells showed higher activation than closed cells near the rat even though all cell types were positively activated (*Figure 3F–H* and *Figure 3—figure supplement 3*). These data agree with reports showing dPAG activation with predator proximity (*Deng et al., 2016*; *Esteban Masferrer et al., 2020*). The present results also showed that proximity to a live predator activated the neither cell ensemble. As the relatively low threat open arms did not significantly activate it, this ensemble of neurons may require a higher degree of threat to be activated relative to open cells.

Closed cells showed increased activity following onset of both escape and freeze, despite these behaviors having opposite motor outputs (*Figure 3I–K*). Additionally, even though freezing and approach onset occur similarly far from the rat (*Figure 3A*), open and closed ensembles showed opposite activity patterns and different generalized linear model (GLM) weights (*Figure 3I–K*). The

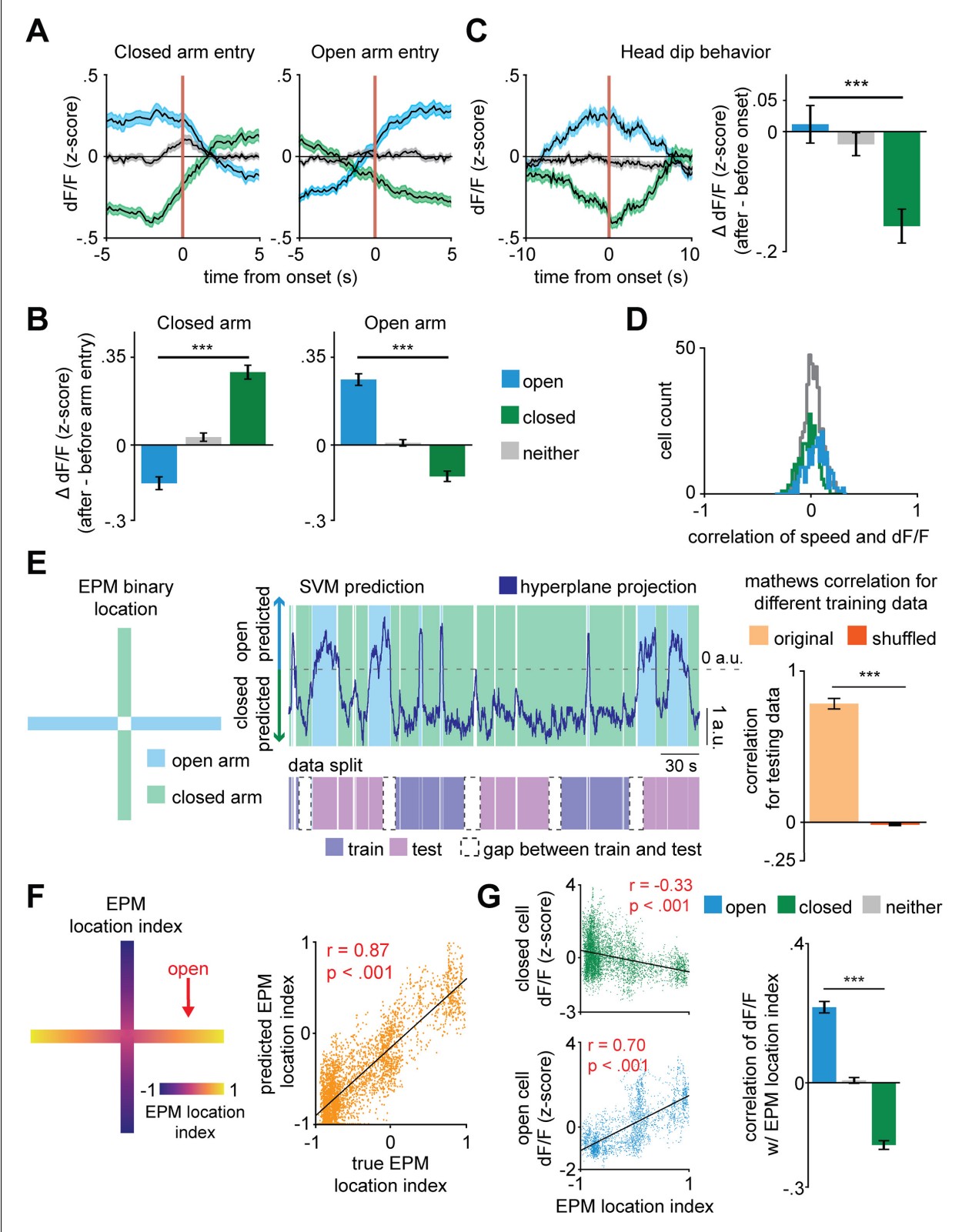

**Figure 2.** Dorsal periaqueductal gray (dPAG) population activity predicts elevated plus maze (EPM) exploration. (**A**) Traces show the mean z-scored activity (±1 SEM) of all open, closed, and neither cells, behavior-aligned to arm transitions (respectively the blue, green, and gray traces). Cell categorization and calculated results were performed on non-overlapping training and testing data (*n* = 180 open cells, *n* = 513 neither cells, *n* = 164 closed cells). (**B**) Bars depict the change in z-scored d*F*/*F* for entries to closed left) and open (right) arms, separately for open, closed, and neither cells

*Figure 2 continued on next page*

*Figure 2 continued*

(data are represented as mean ± SEM; both closed and open arms; *n* of cells same as (**A**); closed arm *U* = 10.6, \*\*\*p<0.001, open arm *U* = 10.8, p<0.001, Wilcoxon rank sum test, *n* = 8 mice). (**C**) Average activity traces for open, closed, and neither cells relative to onset of head dips in the EPM and quantification of changes in activity for all cell types (0–2.5 s after minus 2.5–5.0 s before head dip onset) (data are represented as mean ± SEM; *n* of cells same as (**A**); *U* = 3.9, \*\*\*p<0.001, Wilcoxon rank sum test). (**D**) Histograms depict the distribution of the Pearson's correlation of d*F/F* with speed for each cell type in the EPM (*n* of cells same as *Figure 1I*. (**E**) Prediction of arm-type mouse position in the EPM from neural data using a linear support vector machine (SVM). The blue and green areas represent the actual arm-type occupancy label (open and closed arm, respectively), and the black trace represents the prediction of arm location by the SVM hyperplane projection. If the trace was above 0 a.u., then that period was classified as open arm exploration, otherwise, it was classified as closed arm occupancy. The pink and purple represent the data split (training and testing data, respectively). The Matthews correlation coefficient for real and permuted shuffled training data are shown to the right (mean ±1 SEM; *n* = 8 mice, *U* = 4.9, p<0.001, Wilcoxon rank sum test). (**F**) (Left) Example of EPM location index where 1 and −1 correspond, respectively, to the extreme end of the open and closed arms and (right) prediction of labeled EPM location index from dPAG neural data for an example mouse (scatterplot displays testing data that was not used for training, *r* = Pearson's correlation coefficient). (**G**) (Left) Correlation of example closed and open cell activity and (right) mean correlation of d*F/F* with EPM location index (data are represented as mean ± SEM; *n* of cells same as (**A**); *U* = 13.7, \*\*\*p<0.001, Wilcoxon rank sum test, *r* = Pearson's correlation coefficient).

The online version of this article includes the following figure supplement(s) for figure 2:

**Figure supplement 1.** Validation of linear regression to predict elevated plus maze (EPM) location index using dorsal periaqueductal gray activity.

freezing data in *Figure 3I* excludes freeze bouts that happened within 10 s of an escape. Thus, the activity patterns plotted during freezing cannot be attributed to slow decay induced by escapes. Rat movement onset likely constitutes a threat signal and switches states from approach to avoidance, as predator movement is indicative of increased threat imminence. Indeed, rat movement is a significant predictor variable for less frequent approach and more occurrences of threat avoidance-related behaviors, such as escape and freezing (*Figure 3—figure supplement 4A*). Interestingly, rat movement bouts are not consistently directed toward the mouse, as it most often increases distance toward the mouse (*Figure 3—figure supplement 4B–C*). Nevertheless, rat movements predict escape and freezing, indicating it is perceived as a threat by the mouse. Accordingly, after rat movement onset, the threat avoidance-related closed arm ensembles displayed higher activity (*Figure 3I–J*). Notably, neither of the ensembles consistently resembled the overall average dPAG activity during rat exposure (*Figure 3E*), as each ensemble had its own functional profile (*Figure 3I*). Furthermore, dPAG cells also used shared patterns of neural activity across rat and EPM assays to represent threat imminence (measured as distance to threat) (*Figure 3—figure supplement 5*, see Materials and methods). These results suggest that dPAG neuronal activity can represent internal brain states using shared patterns of activity across different threats.

An approach state is associated with open arm entries, head dips in the EPM, and proximity to threat. Conversely, an avoidance state would be expected far away from threats and during actions that decrease threat exposure, such as closed arm exploration, escape, and freezing. Our results showed that closed cells were more active during higher distance from threat and threat avoidance-related behaviors, such as freezing and escaping, while open cells were more active during proximity to threats and exploratory head dips in the EPM (*Figure 2* and *Figure 3*). The consistency of these results across behaviors and two different threat modalities indicates that dPAG closed and open cells were encoding threat avoidance and threat approach states, respectively.

To further investigate how dPAG cells use a shared representation to encode approach and avoidance states, we developed an avoidance/approach score ranging between −1 and 1 (see Materials and methods). The score gradually increases during approach to threat and during EPM head dips, reaching +1 when the mouse is adjacent to the rat or in the extreme end of the open arms. The score decreases when the mouse is retreating from threat and is assigned a value of −1 when the mouse is furthest from the rat, freezing, or in the extreme end of the closed arms (*Figure 4A*). To investigate how the avoidance/approach score is encoded in dPAG activity, we used *k*-means clustering, an unsupervised approach, to group the data points into clusters (*Figure 4B*, panel 1). We used the Akaike information criterion (AIC) to determine the optimal number of clusters for each assay, which were two clusters for EPM and three for the rat assay (see the Materials and methods section entitled '*k*-Means clustering of neural data'; see also AIC values for a range of cluster numbers in *Figure 4—figure supplement 1*). We then calculated the avoidance/approach score for each cluster (*Figure 4B*, panel 2). The clusters with the lowest and highest scores

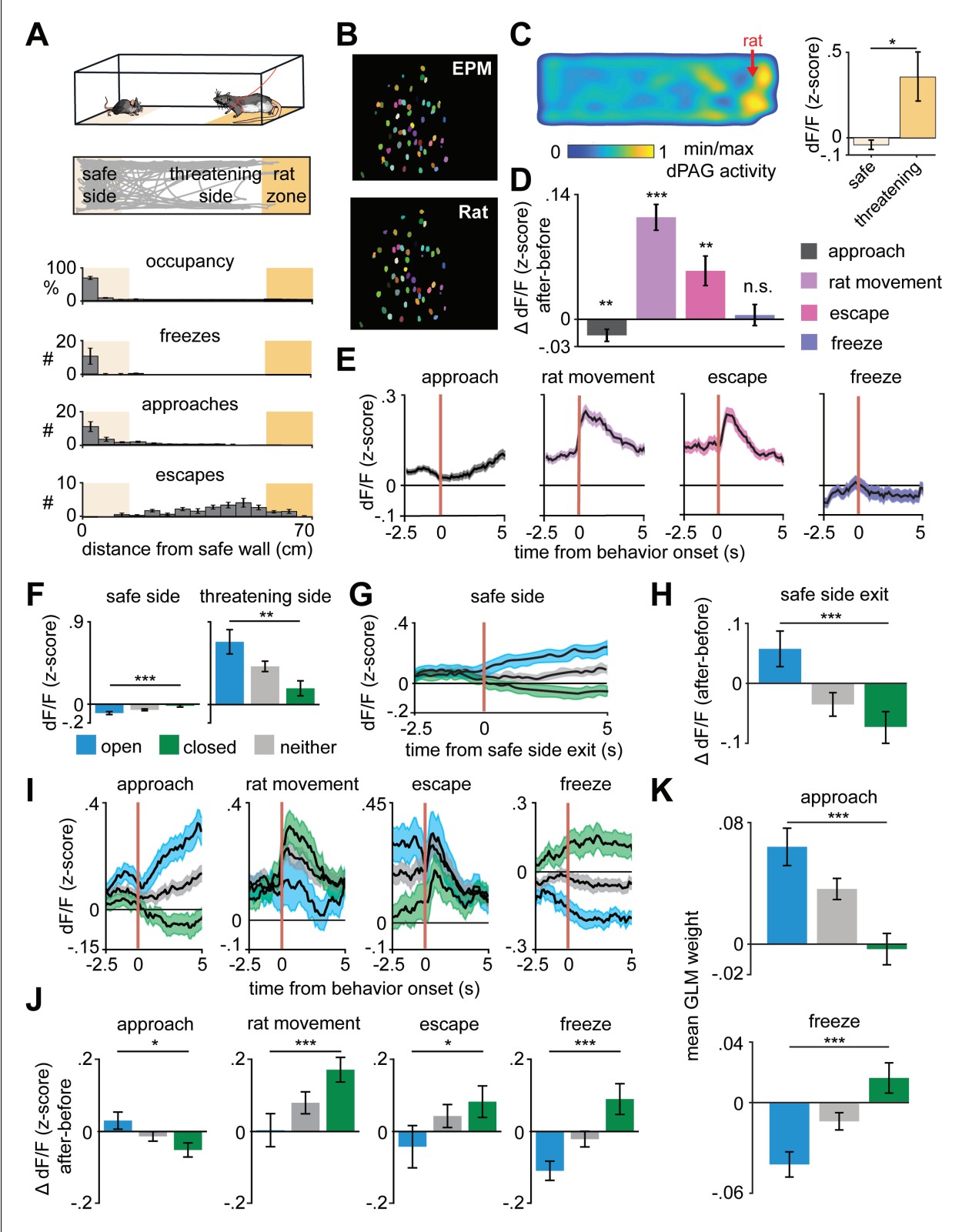

**Figure 3.** Arm-specific ensembles maintain functions across threatening situations. (**A**) Illustration of the rat exposure assay (top) and example track (bottom), with labels depicting the area to which the rat is confined (rat zone) as well as areas near to and far from the rat (safe side). In all figures depicting this assay the rat area will be shown to the right. (**B**) Example imaging field of view with dorsal periaqueductal gray (dPAG) cells co-registered between elevated plus maze (EPM) and rat exposure sessions. (**C**) Heatmap depicts the mean z-scored d*F/F* at each position of the rat exposure assay

*Figure 3 continued on next page*

Figure 3 continued

(left, *n* = 713 cells). Mean d*F/F* is significantly greater in the threatening zone than in the safe zone (*n* = 713 cells, *n* = 7 mice, *U* = 2.24, p=0.03, Wilcoxon rank sum test). (**D**) Change in d*F/F* (0–2.5 s after minus 0–2.5 s before) activity for all dPAG cells for behaviors in the rat exposure assay (data are represented as mean ± SEM; *n* of cells for approach, rat movement, escape = 714, *n* of cells for freeze = 640; approach *t* = −2.65, **p=0.008, rat movement *t* = 7.87, ***p<0.001, escape *t* = 3.28, **p=0.001, freeze *t* = 0.39, p=0.69, one-sample *t*-test). (**E**) Traces show the mean z-scored activity of all cells (±1 SEM), aligned to onset of various behaviors (onset is indicated by the red vertical line) in the rat exposure assay (*n* of cells same as D). (**F**) Bars depict the mean z-scored d*F/F* of cells on the safe side and threatening side of the enclosure (data are represented as mean ± SEM; *n* = 64 open cells, *n* = 166 neither cells, *n* = 87 closed cells; *n* = 7 mice, safe *U* = −3.82, ***p=0.0001, threatening *U* = 3.05, **p=0.002, Wilcoxon rank sum test). (**G**) Traces show the mean z-scored activity of open, closed, and neither cells (±1 SEM), aligned to exit of the safe side of the enclosure (far from the rat). (**H**) Bars show the mean change in z-scored d*F/F* (0–2.5 s after minus 0–2.5 s before) aligned to safe side exit for open, closed, and neither cells (data are represented as mean ± SEM; *n* = 64 open cells, *n* = 166 neither cells, *n* = 87 closed cells; *U* = 3.29, ***p=0.001, Wilcoxon rank sum test). (**I**) Traces show the mean z-scored activity of open, closed, and neither cells (±1 SEM), aligned to behaviors in the rat exposure assay (approach, rat movement, escape: *n* = 64 open cells, *n* = 87 closed cells, *n* = 166 neither cells; freeze: *n* = 50 open cells, *n* = 67 closed cells, *n* = 132 neither cells). Onset of behaviors is indicated by a red vertical line. (**J**) Bars depict the change in z-scored d*F/F* (0–2.5 s after minus 0–2.5 s before) for behaviors in the rat exposure assay, separately for open, closed, and neither cells (data are represented as mean ± SEM; *n* of cells same as (I); approach *U* = 2.45, *p=0.014, rat movement *U* = −3.70, ***p=0.0002, escape *U* = −2.12, *p=0.034, freeze *U* = −3.62, ***p=0.0003, Wilcoxon rank sum test). (**K**) A generalized linear model (GLM) to predict single-cell activity was constructed using approach, escape, and freeze behaviors as variables. Bar plots show average GLM weights for approach and freeze for open, closed, and neither cells (data are represented as mean ± SEM; *n* = 62 open cells, *n* = 155 neither cells, *n* = 83 closed cells; approach *U* = 4.17, ***p<0.001, p=0.11, freeze *U* = 3.02, ***p<0.0002, Wilcoxon rank sum test).

The online version of this article includes the following figure supplement(s) for figure 3:

**Figure supplement 1.** Validation of co-registration procedure.
**Figure supplement 2.** Behavioral and dorsal periaqueductal gray (dPAG) activity differences between rat and toy rat exposure.
**Figure supplement 3.** Correlation of dorsal periaqueductal gray cell ensembles with distance to a rat.
**Figure supplement 4.** Increased rat velocity predicts lower approach to rat and higher threat avoidance-related behaviors such as escape and freeze.
**Figure supplement 5.** Dorsal periaqueductal gray (dPAG) displays a shared neural representation of risk imminence across the elevated plus maze (EPM) and rat exposure assays.

were classified as the 'avoidance' and the 'approach' cluster, respectively. Experimentally observed approach and avoidance clusters respectively had higher and lower scores than sampling distributions obtained from permuted behavioral labels, showing that our *k*-means approach identified activity patterns that strongly encode the avoidance/approach score (*Figure 4C*). The approach and avoidance cluster centroids from one assay were then applied to the other assay (i.e., centroids were defined by training on EPM and applied on previously unseen data from the rat assay, or vice versa). Cluster centroids defined from the training data in one assay were applied to the data from the other assay to assign cluster identity based on shortest Euclidean distance to the centroid. For example, the points in the testing dataset that were closest to the avoidance centroid, previously defined by the training dataset, were assigned to the avoidance cluster (*Figure 4B*, panels 3–4). Scores for approach and avoidance clusters for shuffled data were used to create a sampling distribution. Lastly, we show that approach and avoidance clusters trained on one assay and applied on the other assay result in significantly different scores, despite the two assays having different geometries and distinct threat modalities (*Figure 4D*). Importantly, these results were not found when computing approach and avoidance cluster centroids defined on the control toy rat assay but applied to the rat assay (*Figure 4—figure supplement 2*). These results indicate, using an unsupervised method, that approach and avoidance states are encoded using shared patterns of neural activity across assays. Similar results were also obtained by employing a hidden Markov model (HMM), showing that the results in *Figure 4* can be found using a diversity of computational approaches (*Figure 4—figure supplement 3*). Importantly, dPAG activity reflects moment-to-moment changes in the behavioral states of the animal. Rather than merely encoding the defensive behavior movements, these activity patterns may indicate threat perception. Encoding of exploratory and defensive behavior has also been reported in the amygdala (*Gründemann et al., 2019*), indicating that multiple defensive nodes represent such brain states.

Finally, we investigated if ensemble composition was related to threat avoidance traits across assays. Rat approach and open arm exploration were correlated across mice, indicating that these measures also reflected trait avoidance levels (*Figure 4—figure supplement 4A*). We then found that mice with a higher proportion of open cells in relation to closed cells displayed increased avoidance of open arms and rat (*Figure 4—figure supplement 4B–C*). A possible interpretation for these

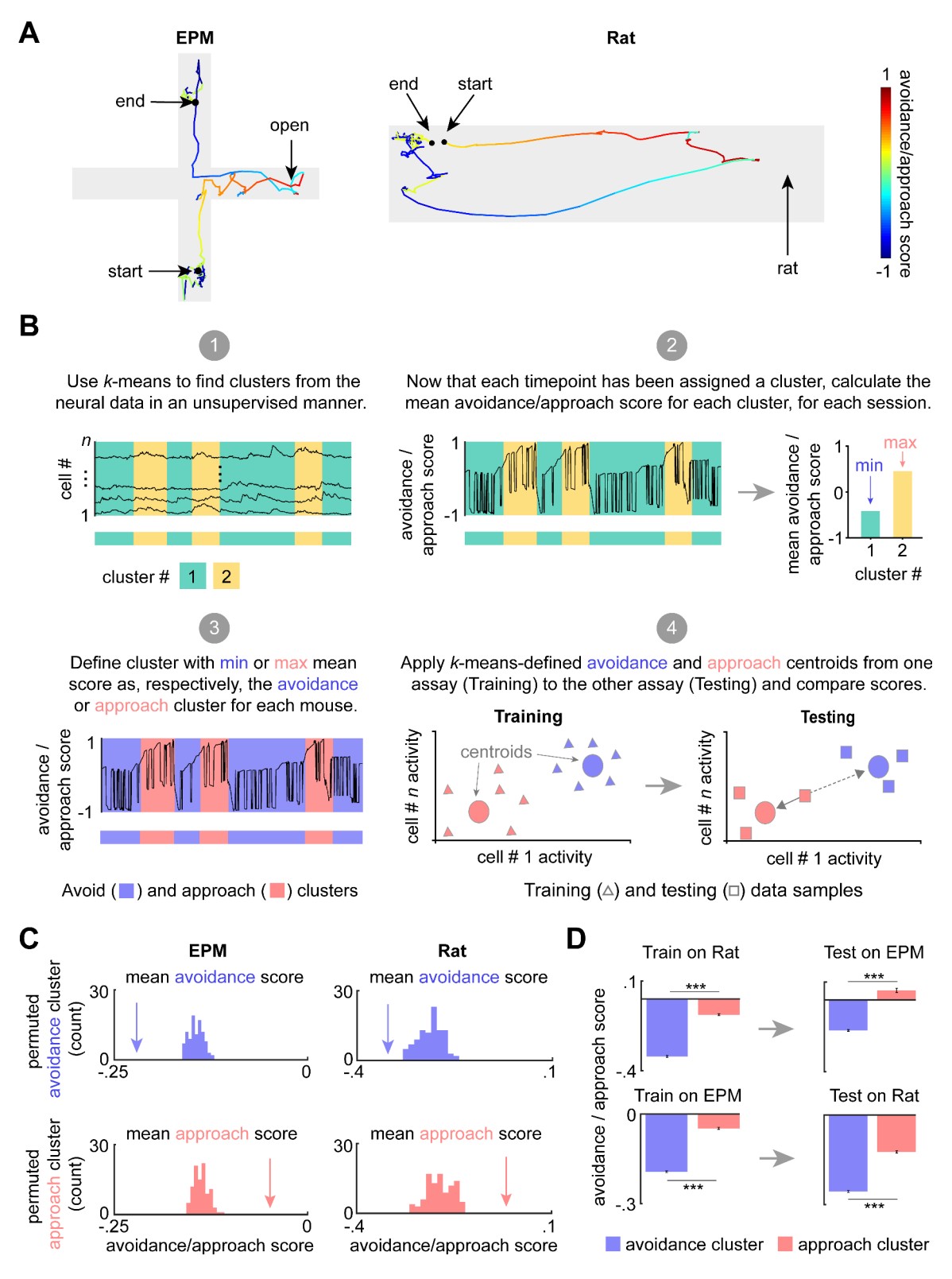

**Figure 4.** Dorsal periaqueductal gray (dPAG) displays a shared neural representation of avoidance and approach states across the elevated plus maze (EPM) and rat exposure assays. (**A**) Example tracks in the EPM (left) and rat exposure assay (right), color-coded by avoidance/approach score (see Materials and methods). The approach score increased during movements toward the threat, reaching +1 when the animal reaches the end of the open arms or the rat. The score decreases during movement away from threat and reaches its minimum value of −1 when the mouse reaches the end of

*Figure 4 continued on next page*

*Figure 4 continued*

closed arms or the furthest point from the rat. This score was developed as a measure of avoidance/approach states. (B) Explanatory diagram depicting steps of the clustering analysis (see Materials and methods). (1) *k*-Means was used to find clusters in the neural data in an unsupervised manner. (2) The mean avoidance/approach score was calculated for each cluster defined in step 1. (3) The 'avoidance' and 'approach' clusters were identified as those with, respectively, the minimum or maximum mean avoidance/approach score calculated in step 2. (4) The approach and avoidance centroids defined in one assay were used to classify neural data from the other assay, based on the minimum Euclidean distance for each sample (as depicted by solid arrow). (C) Arrow depicts the experimentally observed mean avoidance/approach score for avoidance and approach clusters (EPM $k = 2$; rat $k = 3$, see *Figure 4—figure supplement 1*) across concatenated sessions ($n = 7$ mice). This mean was compared to a distribution of avoidance (top) or approach (bottom) cluster means, calculated by permuting the neural data 100 times (EPM cells $n = 734$, rat assay cells $n = 713$; for all, p<0.01). (D) (Left) Bars depict the mean rat and EPM avoidance/approach scores ($\pm 1$ SEM) for approach and avoidance clusters across mice. (Right) As described in Materials and methods and Figure 4B, these cluster centroid locations, trained on one assay, were then used to define approach and avoidance timepoints in the other assay. Bars depict the corresponding mean avoidance/approach score ($\pm 1$ SEM) for this testing data (train on EPM: avoidance cluster $n = 31,938$, approach cluster $n = 22,630$; test on rat: avoidance cluster $n = 30,245$, approach cluster $n = 24,412$; train on rat: avoidance cluster $n = 14,658$, approach cluster $n = 10,514$; test on EPM: avoidance cluster $n = 15,319$, approach cluster $n = 2510$ ($n$ represents the number of timepoints, not cells); co-registered cells $n = 317$; Wilcoxon rank sum test, ***p<0.001).

The online version of this article includes the following figure supplement(s) for figure 4:

**Figure supplement 1.** Determination of optimal number of clusters for elevated plus maze (EPM) and rat assay according to Akaike information criterion (AIC).

**Figure supplement 2.** Dorsal periaqueductal gray (dPAG) ensembles do not encode avoidance and approach to a control toy rat.

**Figure supplement 3.** Hidden Markov models also differentiate avoidance and approach states.

**Figure supplement 4.** Fraction of open arm cells was negatively correlated with approach to threat across mice.

---

results is that, rather than causing open arm exploration, the open arm ensemble is highly responsive to risk assessment and proximity to threat. Considering this hypothesis, our results suggest that mice with more open arm cells display an increased sensitivity to risk evaluation reflected on increased avoidance of threat. These data indicate that in addition to encoding moment-to-moment changes in behavioral state, dPAG ensembles composition may integrate risk evaluation processes and influence individual mouse differences in threat avoidance traits.

The dPAG's role in defense has been described as restricted to the initiation of escape (*Evans et al., 2018*). A proposed model is that excitatory neurons initiate escape (but not other defensive behaviors) in response to imminent threat (*Evans et al., 2018*). Accordingly, optogenetically stimulating vglut2-positive cells in dPAG caused escape and presenting looming stimuli while optogenetically inhibiting those cells caused freezing (*Evans et al., 2018*). Thus, threat information would be transmitted to the dPAG, resulting in initiation of escape and inhibition of freezing (*Lefler et al., 2020*; *Tovote et al., 2016*). However, recent data indicate that the dPAG's influence on defense is not restricted to escape initiation. For example, optogenetic activation of dPAG CaMKIIα-positive neurons can cause both escape and freezing (*Deng et al., 2016*). Increasing the optogenetic stimulation frequency of CaMKIIα-positive dPAG neurons switches responses from freezing to escape (*Deng et al., 2016*). Supporting the view that the dPAG may promote freezing, activation of excitatory ventromedial hypothalamus projections to the dPAG induces freezing (*Wang et al., 2015*). Conversely, the hypothalamic dorsal premammillary nucleus projections to the dPAG control escape elicited by numerous innate threats (*Wang et al., 2021*). Recent work shows that the medial prefrontal cortex promotes aversion and might regulate defensive strategies through projections to the dPAG (*Vander Weele et al., 2018*). These findings indicate that through different inputs, the dPAG may integrate distinct aspects of emotional states, detect diverse threats, and coordinate a range of defensive responses, including freezing and escape.

We propose that when the subject can voluntarily approach the threat, the dPAG has a broader function in threat processing. It may then function as a risk estimator that represents approach- and avoidance-related states, and consequently influences a wide range of behaviors related to both approach and avoidance. In contrast, there are situations in which threats appear independently of voluntary approach, such as during the presentation of looming stimuli. In such cases, approach cannot be assessed as only avoidance behaviors such as freezing and escape are possible (*Evans et al., 2018*; *Salay et al., 2018*). Hence, these situations only allow assessing the dPAG's involvement in avoidance, but not approach-related states.

Our study reveals novel features of dPAG neurons. First, dPAG cells differentiated between open and closed spaces in the EPM. Second, we show that closed cell activity patterns were positively

correlated with threat perception represented by the predator's movement. Furthermore, despite having opposite motor outputs, escape and freezing correlated with increased closed cell activity. Conversely, open cell activity was inversely correlated with distance to threat. Closed and open cells show positive z-scored activity following escape onset. However, open cell activity starts decreasing following escape onset, while closed cell activity increases after flight initiation (*Figure 3J*). Thus, our data highlight the importance of considering activity dynamics rather than only analyzing if cells display high or low activity during a behavior. Our observations suggest that dPAG is not simply a premotor area but rather reflects the subjects' underlying emotional state, in agreement with previous work (*Carvalho et al., 2015*; *Carvalho et al., 2018*; *Johansen et al., 2010*; *Kim et al., 2013*; *Kincheski et al., 2012*; *Nashold et al., 1969*). This idea is consistent with the fact that dPAG activity represented location within the EPM (*Figure 2F*), and that the dPAG used shared patterns of neural activity across rat and EPM assays to represent threat imminence (*Figure 3—figure supplement 5*) and behavioral states (*Figure 4*). The finding that the dPAG uses a conserved representation of threat approach and avoidance across assays expands prior results using stimulation, calcium imaging, electrophysiology, and c-fos expression (*Bittencourt et al., 2005*; *Brandão et al., 1982*; *Canteras and Goto, 1999*; *Deng et al., 2016*; *Esteban Masferrer et al., 2020*; *Evans et al., 2018*). A role beyond motor output execution has also been proposed for the ventral PAG, as its neural activity reflects threat probability associated with specific cues (*Wright and McDannald, 2019*; *Wright et al., 2019*).

A possible mechanistic explanation for the differences between open and closed ensembles is that they might correspond to specific neural populations integrating different neurotransmitter systems with particular modulatory effects over defensive behaviors (e.g., glutamate, GABA, substance P, nitric oxide, etc.). Alternatively, these ensembles may have distinct input connectivity, leading to distinct activity patterns. Based on its connectivity, the dPAG is well positioned to have a privileged role in computing threat imminence and threat probability, as it receives input from sensory, limbic, and cognitive areas (*Silva and McNaughton, 2019*), and projects not only to centers that control motor actions (*Ferreira-Pinto et al., 2018*; *Marchand and Hagino, 1983*) but also to prosencephalic targets likely to mediate fear behavior (*Gross and Canteras, 2012*; *Krout and Loewy, 2000*; *Motta et al., 2017*). Future studies are needed to further dissect dPAG cell types based on genetic markers, connectivity, and functional differences. In summary, our study reveals two large functional neuronal ensembles of the dPAG representing internal states. By performing population-level analysis of dPAG neurons in two different threatening situations, we were able to demonstrate that dPAG cells are not only encoding specific behaviors or threat imminence (*Deng et al., 2016*; *Esteban Masferrer et al., 2020*; *Evans et al., 2018*; *Watson et al., 2016*), but also that dPAG ensemble activity reflects moment-to-moment changes in the approach-avoidance state of the animal. These findings expand on the oversimplified view of dPAG as a premotor output region and highlight it as a key node reflecting the internal brain states that prepare the organism to engage in approach or avoidance of threat.

# Materials and methods

**Key resources table**

| Reagent type (species) or resource | Designation | Source or reference | Identifiers | Additional information |
|---|---|---|---|---|
| Strain, strain background (*Mus musculus*) | C57BL/6J | Jackson Labs | RRID:IMSR_JAX:000664 | |
| Strain, strain background (*Rattus norvegicus*) | Long-Evans | Charles River Labs | RRID:RGD-631593 | |
| Recombinant DNA reagent | AAV9.Syn.GCaMP6s.WPRE.SV40 | Addgene | 100843-AAV9 | |
| Software, algorithm | MATLAB | Mathworks | RRID:SCR_001622 | |

## Mice

Mice (*Mus musculus*) of the C57BL/6J strain (Jackson Laboratory stock No. 000664) were used for all experiments. Male mice between 2 and 5 months of age were used in all experiments. Mice were maintained on a 12 hr reverse light-dark cycle with food and water ad libitum. Sample sizes were

chosen based on previous behavioral studies with miniaturized microscope recordings on defensive behaviors, which typically use 6–10 mice per group. All mice were handled for a minimum of 5 days prior to any behavioral task. In this work, analyses of the EPM environment used eight mice, while any analyses involving rat exposure used seven mice. Sample size was chosen based on prior dPAG calcium transient recordings (*Evans et al., 2018*). All procedures have been approved by the University of California, Los Angeles Institutional Animal Care and Use Committee, protocols 2017–011 and 2017–075.

### Rats

Male Long-Evans rats (250–400 g) were obtained from Charles River and were individually housed on a standard 12 hr light-dark cycle and given food and water ad libitum. Rats were only used as a predatory stimulus. Rats were handled for several weeks prior to being used and were screened for low aggression to avoid attacks on mice. No attacks on mice were observed in this experiment.

### Surgeries

Eight-week-old mice were anaesthetized with 1.5–3.0% isoflurane and placed in a stereotaxic apparatus (Kopf Instruments). AAV9.Syn.GCaMP6s.WPRE.SV40 were packaged and supplied by UPenn Vector Core at titers $7.5 \times 10^{13}$ viral particles per ml and viral aliquots were diluted prior to use with artificial cortex buffer to a final titer of $5 \times 10^{12}$ viral particles per ml. After performing a craniotomy, 100 nl of virus was injected into the dPAG (coordinates in mm, from skull surface): −4.20 anteromedial, −0.85 lateral, −2.3 depth, 15-degree angle. Five days after virus injection, the animals underwent a second surgery in which two skull screws were inserted and a microendoscope was implanted above the injection site. A 0.5 mm diameter, ~4-mm-long gradient refractive index (GRIN) lens (Inscopix, Palo Alto, CA) was implanted above the dPAG (−2.0 mm ventral to the skull surface) (*Resendez et al., 2016*). The lens was fixed to the skull with cyanoacrylate glue and adhesive cement (Metabond; Parkell, Edgewood, NY, USA). The exposed end of the GRIN lens was protected with transparent Kwik-seal glue and animals were returned to a clean cage. Two weeks later, a small aluminum base plate was cemented onto the animal's head on top of the previously formed dental cement. Animals were provided with analgesic and anti-inflammatory (carprofen).

### Behavioral timeline

Behavioral tests were combined in the following manner across days: EPM test, rat exposure environment habituation (no rat), toy rat exposure, and rat exposure. Each mouse was only exposed to each assay once, as fear assays cannot be repeated. Thus, there are no technical replicates. No outliers were found or excluded. Neural recordings were obtained from all mice in identical conditions, and thus they were all allocated to the same experimental group. There were no experimentally controlled differences across mice and there were no 'treatment groups'.

### EPM test

Mice were placed in the center of the EPM facing one of the closed arms and could freely explore the environment for 20 min. The length of each arm was 30 cm, the width was 7 cm, and the height of the closed arm walls was 20 cm. The maze was 65 cm elevated from the floor by a camera stand. A total of eight mice were analyzed.

### Rat exposure assay

Twenty-four hours after the EPM test, mice were habituated to a white rectangular box (70 cm length, 26 cm width, 44 cm height) for a 20 min session. On the following day, mice were exposed to the same environment for 20 min but in the presence of a toy rat. Twenty-four hours later, mice were exposed to an adult rat in this environment. The rat was secured by a harness tied to one of the walls and could freely ambulate only within a short perimeter. The mouse was placed near the wall opposite to the rat and freely explored the environment for 20 min. No separating barrier was placed between the mouse and the rat, allowing for close naturalistic encounters that can induce a variety of robust defensive behaviors. One mouse was removed from the analysis due to poor recording quality. This is why a total of seven mice were analyzed, instead of eight like in the EPM.

### Behavior and miniscope video capture

All videos were recorded at 30 frames/s using a Logitech HD C310 webcam and custom-built head-mounted UCLA miniscope (*Aharoni and Hoogland, 2019*). Open-source UCLA miniscope software and hardware (http://miniscope.org/) were used to capture and synchronize neural and behavioral video (*Cai et al., 2016*; *Schuette et al., 2020*). Neural recordings were sampled at 7.5 Hz.

### Perfusion and histological verification

Mice were anesthetized with Fatal-Plus and transcardially perfused with phosphate-buffered saline followed by a solution of 4% paraformaldehyde. Extracted brains were stored for 12 hr at 4°C in 4% paraformaldehyde. Brains were then placed in sucrose solution for a minimum of 24 hr. Brains were sectioned in the coronal plane in a cryostat, washed in phosphate-buffered saline, and mounted on glass slides using PVA-DABCO. Images were acquired using a Keyence BZ-X fluorescence microscope with a 10 or 20× air objective.

Data Analysis was performed using custom-written code in MATLAB and Python.

### Miniscope postprocessing and co-registration

Miniscope videos were motion-corrected using the open-source UCLA miniscope analysis package (https://github.com/daharoni/Miniscope_Analysis) (*Aharoni and Hoogland, 2019*). They were spatially downsampled by a factor of 2 and temporally downsampled by a factor of 4, and the cell footprints and activity were extracted using the open-source package Constrained Nonnegative Matrix Factorization for microEndoscopic data (CNMF-E; https://github.com/zhoupc/CNMF_E) (*Zhou et al., 2018*). Neurons were co-registered across sessions using the open-source probabilistic modeling package CellReg (https://github.com/zivlab/CellReg) (*Sheintuch et al., 2017*). On average, 45.3 ± 19.4 cells (43.6 ± 15.9%) were co-registered across EPM and rat exposure assays for the seven mice analyzed for both assays.

### Artifact suppression

For suppression of long time scale artifacts, for example, long time scale fluctuations in calcium fluorescence shared across many neurons due to bleaching or other factors, we used PCA (principal component analysis) to identify large variance principal components (PCs) (≥5% total variance) where the projected data exhibited artifacts. Artifacts are typically large in magnitude and occur across many neurons, resulting in dimensions of high variance that resemble artifacts. Cell activity was then reconstructed by excluding these PCs from reconstruction (*O'Shea and Shenoy, 2018*). This method was applied only to data for mouse 1 in the rat exposure assay.

### Variance thresholding

A minority of recorded candidate cells had very small variance over the course of an experimental session. To exclude these cells from analysis, we identified a reference cell for each trial. To choose the reference cell, cell variances were first sorted in decreasing order and then plotted. By visual inspection, the cell at the elbow of the plot was chosen as the reference cell. If there was no obvious elbow, the cell with the highest variance was chosen. Candidate cells with less than 10% of the reference cell's variance were discarded. The remaining cells were z-scored and used for further analysis. Among the 375 total coregistered cells, 10 were excluded because of low variance in both the EPM and Rat, 11 were excluded because of low variance in the EPM but were not excluded in the Rat, and 38 were excluded in the Rat but not the EPM.

### z-Scoring of activity

Cell activity was z-scored once for the whole trace prior to use of the neural activity data in any analysis. We denote the cell activity as $x$, and the mean and standard deviation of the cell's activity over the entire experimental session as $\mu$ and $\sigma$, respectively. The z-scored cell activity was then computed as $(x - \mu)/\sigma$.

### Behavior detection

To extract the pose of freely behaving mice in the described assays, we implemented DeepLabCut (*Mathis et al., 2018*), an open-source convolutional neural network-based toolbox, to identify mouse

nose, ear, and tailbase *xy*-coordinates in each recorded video frame. These coordinates were then used to calculate velocity and position at each timepoint, as well as classify defensive behaviors in an automated manner using custom MATLAB scripts. Freezing was defined as epochs when head and tailbase velocities fell below 0.25 cm/s for a period of 0.33 s. Approach and escape were defined as epochs when the mouse moved, respectively, toward or away from the rat at a velocity exceeding a minimum threshold of 3 cm/s.

We also measured the onset of prominent rat movements, as predator motion elicited defensive behavior in the mouse (*Figure 3—figure supplement 4A*). We identified all rat movements in which rat motion was above the 99.5th percentile of rat speed during the session. This approach ensured that only large predator movements were used, rather than minor changes in position, such as small rat tail movements. If multiple rat movements occurred within 5.33 s of each other, only the first was used for analysis. Rat movement direction was measured with respect to the position of the mouse at movement initiation. A 0-degree movement was toward the mouse's initial position and a 180-degree movement was away from the mouse's initial position; 90- and 270-degree movements were sideways from the mouse's initial position.

## Categorization of open, neither, and closed arm-preferring cells

A cell was categorized as an open arm-preferring cell if activity in each individual open arm was significantly greater than the pooled activity in the closed arms (Wilcoxon rank sum test, p<0.05). Likewise, a closed arm-preferring cell was identified as a cell whose activity in each individual closed arm was significantly greater than the pooled activity in the open arms. The remaining cells were labeled as neither arm-preferring cells. The activity and correlations of open, closed, and neither arm cells in *Figure 2A–C,G* were computed on withheld data. The first 40% of each session was used for cell categorization while the last 60%, after a 10 s separation, was used to calculate activity traces and correlations. All other results used data from the whole session for cell categorization.

## Behavior-aligned trace and Δd*F*/*F* activity

We calculated each cell's behavior-aligned activity by computing the mean activity of the cell over all behavior occurrences, aligned to behavior onset. The mean peri-behavior trace for an ensemble (e.g., closed cells, open cells, or neither cells) was the average of peri-behavior activity across all cells in the ensemble. Change in mean activity after and before behavior was calculated by first subtracting the mean activity of each cell during the time frame $[-2.5, 0]$ seconds relative to behavior onset from the mean activity of each cell in the time frame $[0, 2.5]$ seconds. The overall difference in an ensemble, denoted Δd*F*/*F*, was the average of the change in mean activity across all cells in the ensemble. For head dips, Δd*F*/*F* was calculated using windows of $[-5, -2.5]$ (before) and $[0, 2.5]$ (after). In *Figure 3*, escapes and freezes which occurred within 10 s after the other behavior were excluded to reduce confounding due to slow GCaMP decay.

## Interleaved training and testing data

For analyses involving regression (EPM location index prediction, constrained correlation analysis [CoCA]), as well as SVM classification, testing data were interleaved with training data, with 60 s for each segment and 10 s of separation between data types, that is, [60 s training, 10 s excluded, 60 s testing, 10 s excluded, 60 s training, etc.]. We included gaps to minimize overlapping activity in the training and testing sets, which may arise due to dynamics in calcium activity.

## Open arm preference score

Cells with strong preference for higher activity in the closed arms or open arms respectively have open arm preference scores near $-1$ and $+1$. The arm score quantifies the separability of cell activity between arm types and is invariant to shifting and scaling of the cell activity. Excluding times when the mouse was in the center of the EPM, data points were labeled according to whether the mouse was in an open arm (positive label) or a closed arm (negative label). The open arm preference score was then defined as:

$$\text{Open arm preference score} = 2^*\text{AUC} - 1$$

where AUC is the area under the receiver operating characteristic (ROC) curve resulting from

predicting which arm the mouse was in from single-cell activity. Predictions were made by thresholding cell activity. Cell activity greater than or equal to the threshold was predicted as being in the open arms while cell activity less than the threshold was predicted as the closed arms. We calculated the true positive (open) and false positive (open) rates by varying the threshold across every value of cell activity. These were then sorted to build the ROC curve and calculate the AUC. An open arm preference score of exactly +1 indicates that cell activity in the open arms is strictly greater than activity in the closed arms.

## EPM score

The EPM score (*Adhikari et al., 2011*) quantifies how differently a cell fires between closed vs. open arms. It is close to one when the cell has large differences in activity between arms of different types but is negative if the cell's activity is more similar between different arm types than between same arm types. To calculate EPM score, we first compute the mean difference in z-scored activity between arms of different types (*A*) and arms of the same type (*B*). These are defined as:

$$A = 0.25^*(|F_{C1} - F_{O1}| + |F_{C1} - F_{O2}| + |F_{C2} - F_{O1}| + |F_{C2} - F_{O2}|)$$

$$B = 0.5^*(|F_{C1} - F_{C2}| + |F_{O1} - F_{O2}|),$$

where $F_{O1}$ and $F_{O2}$ are the mean z-scored activity of the cell in open arms 1 and 2, respectively, and $F_{C1}$ and $F_{C2}$ are the mean z-scored activity in closed arms 1 and 2, respectively. The EPM score is defined as:

$$\text{EPM score} = (A - B)/(A + B).$$

Cells with high EPM score if they have large differences in activity in different arm types (large *A*) and similar activity in same type arms (small *B*). The maximum score of 1.0 indicates no difference in cell activity across arms of the same type (*B* = 0). Cells with negative EPM scores have more similar activity across arms of different types than across arms of the same type.

## EPM location index

The EPM location index quantifies the position of the mouse in the EPM. It is close to 1 when the mouse is at the end of an open arm, and close to −1 when the mouse is at the end of the closed arms. To calculate the EPM location index, we first normalized the *x* and *y* position of the mice in the EPM to be in the range [−1, 1], where the *x* position is ±1 at the ends of the open arms and the *y* position is ±1 at the ends of the closed arms. We defined the EPM location index as:

$$\text{EPM location index} = |x| - |y|.$$

The EPM location index is thus continuously varying between −1 and 1. Prediction of EPM location index (*Figure 2F–G*) was performed using linear regression with interleaved training and testing data. Outputs were clipped to the range [−1, 1] before final prediction was made.

Prediction of mouse position in the EPM from neural data using a linear SVM.

After z-scoring data, times when the mouse was in the center of the EPM were removed from training. The remaining data were separated into alternating blocks of 60 s training data and 60 s testing data with 10 s of separation between blocks. A linear SVM was fit on training using scikit-learn's SVC function with balanced class weights (*Pedregosa et al., 2011*). Significance testing was performed using permutation tests with shuffled training labels for 100 random trials per mouse. The Matthews correlation coefficient was used to quantify the relation between predicted and observed arm-type occupancy because this metric was developed to assess correlations between binary values (such as arm type, which can only be closed or open arms).

## Zones in the rat assay

The safe zone was defined as the left 20% of the rat environment, based on *x* position. A mouse was labeled as being in the threatening zone whenever the distance to the rat was 14 cm or less.

## Generalized linear model

A GLM was fit for each cell. Each GLM mapped behavior variables to the cell's z-scored calcium activity. In total, there were eight behavior variables for rat: distance to rat, mouse velocity, rat velocity, angle from mouse's head to the rat, and occurrence of approaching, escaping, and freezing. Discrete behaviors were binary, labeled as one at all times in which they occurred and 0 otherwise. To enable behaviors to alter neural activity prior to and following the behavior, each binarized behavior was convolved with a log-time scaled raised cosine basis, from 5 s before behavior onset to 5 s after behavior offset. This kernel had the form:

$$f_j(x) = \frac{1}{2}\left[\cos\left(\pi x - \varphi_j\right) + 1\right], \frac{1}{\pi}\varphi_j - 1 < x \le \frac{1}{\pi}\varphi_j + 1$$

$$x = \log(t + b + \varepsilon)$$

$$\varphi_j = \log(b + \varepsilon) + \frac{1}{2}(j - 1)\pi$$

These kernels are non-zero when $x$ is between $\left[\frac{1}{\pi}\varphi_j - 1, \frac{1}{\pi}\varphi_j + 1\right]$, and are zero otherwise. Further, to enable historical kinematics to affect present neural data, the kinematics were convolved with a kernel, which used the same set of bases as the behavior, but only had responses after the onset within 5 s. These convolved behavior variables, denoted here as $y_1$, $y_2$, etc., were then modeled to produce the cell's calcium fluorescence as:

$$x = \beta_1 y_1 + \beta_2 y_2 + ... + \beta_m y_m + c,$$

where $\beta_i$ is the coefficient for the $i$th behavior variable. The GLM was optimized by minimizing the mean-square error of the reconstruction between the GLM activity estimate, $x$, and the recorded calcium activity.

## Cross-assay constrained correlation analysis

To investigate if the dPAG uses shared patterns of neural activity to represent threat imminence across assays, we developed Constrained Correlation Analysis (CoCA). This model identifies the features used in a shared neural representation between different threatening situations. The CoCA technique defined a shared neural projection as a linear combination of the activity of individual neurons. Additionally, we constructed a behavioral projection for each assay, through a linear combination of each assay's behavioral variables. Optimization via CoCA produced neural projection weights that were compared across open and closed cell ensembles.

We denote calcium fluorescence neural data as $X \in R^{k \times T}$ and externally observed behavioral data as $Y \in R^{p \times T}$, where $k$ is the number of recorded cells for the corresponding mouse, shared across assays, $p$ is the chosen number of behavioral variables for the corresponding assay, shared across mice, and $T$ is the length of a recording session, unique for each session, but shared between neural and behavioral data for the same session. Behavioral variables contained both continuous kinematic variables (such as speed and distance from rat) and binary defensive behavior variables (such as the occurrence of freezing and escape). All variables were normalized to 0 mean and unit variance, except normalized $|x|$ and $|y|$ position in the EPM, which were instead in the range $[0, 1]$ (variance $\le 0.25$).

In order to find a common linear projection of threat across mice and assays, we performed the following optimization with mouse IDs $i = 1, 2,..., 7$ and assay IDs $j$ = EPM, RAT. Calcium fluorescence traces of dPAG cells for mouse $i$ were linearly combined after multiplying each cell with weights $n_1^i$ to $n_k^i$, where $k$ is the number of cells that were co-registered in both assays. Taking the dot product of the calcium activity for mouse $i$ in assay $j$, given by $X_{i,j}$, and the weights $\mathbf{n}^i = [n_{1...k}^i]$ defined a neural projection for mouse $i$ and assay $j$, given by $N_{i,j} = (\mathbf{n}^i)^T X_{i,j}$, neural data projection in red). For each mouse, the weights, $\mathbf{n}^i$, were the same across assays, so that each cell had the same weight in both assays. The behavioral variables for the EPM (such as $x$ and $y$ position, speed, etc.) were linearly combined with a set of weights $b_1$ to $b_6$ (as six behavioral variables were used for the EPM). These weights, $\mathbf{b}^{EPM} = [b_1^{EPM}, b_2^{EPM}, ..., b_6^{EPM}]$ were conserved across all mice. Linearly combining the EPM behavioral variables resulted in a behavioral projection for mouse $i$ and assay EPM,

given by $B_{i,EPM} = (\mathbf{b}^{EPM})^T Y_{i,EPM}$. Similarly, nine behavioral variables from the rat assay were linearly combined to produce a behavioral projection $B_{i,RAT} = (\mathbf{b}^{RAT})^T Y_{i,RAT}$ using weights $\mathbf{b}^{RAT} = [b_1^{RAT}, b_2^{RAT}, \ldots, b_9^{RAT}]$. We chose the neural weights, $\mathbf{n}^i$, and the behavioral weights, $\mathbf{b}^{EPM}$ and $\mathbf{b}^{RAT}$, to optimize the correlations across all mice and assays:

$$\max \sum_i \sum_j \mathrm{corr}\left(N_{i,j}, B_{i,j}\right)$$

where corr() is the Pearson's correlation coefficient, and $N_{i,j}$ and $B_{i,j}$ are the linear projections of neural and behavioral data, respectively, given by $N_{i,j} = (\mathbf{n}^i)^T X_{i,j}$ and $B_{i,j} = (\mathbf{b}^j)^T Y_{i,j}$. The optimization variables $\mathbf{n}^i$, $i = 1, 2, \ldots, 7$ and $\mathbf{b}^j$, $j = $ EPM, RAT were simultaneously optimized using gradient descent via the Adam optimizer (*Kingma and Ba, 2015*) until convergence. Results presented use interleaved training and testing data. This method was implemented using PyTorch.

## CoCA significance testing

In order to test if correlations of testing data were better than expected by chance, correlations were computed between projected behavioral data (using weights fit by training data) and random projections of neural data (1000 trials). We emphasize that these correlations were applied to the testing data, and therefore it was possible for a random projection to have higher correlation than the CoCA projection. A one-tailed test was used (p<0.05).

## Avoidance/approach score

To calculate the continuous avoidance/approach score for each assay, the distance from safety was calculated (rat: distance from the safe wall; EPM: distance from the end of the closed arm) and normalized such that it ranged from 0 to 1 in the rat assay and 0 to 0.9 in the EPM assay. A binarized direction value was also assigned to each timepoint, indicating if the mouse was moving toward (+1) or away from (−1) the threat. To incorporate categorized behaviors, the avoidance/approach score for freeze samples equaled the minimum score of −1. For the EPM only, the avoidance/approach score was multiplied by 1.11 for head dip samples, such that a head dip at the end of the open arm would yield the maximum score of 1.

To calculate the score at each timepoint:

> While approaching threat, avoidance/approach score = distance to safety × direction
> While avoiding threat: avoidance/approach score = [1−distance to safety] × direction

## *k*-Means clustering of neural data

To determine if the avoidance/approach score is represented in the neural data, the *k*-means algorithm was used to cluster this data in an unsupervised manner. One of the simplest unsupervised classification algorithms, this approach identifies groupings of activity that are strongly represented in the neural data, without providing any additional information about animal behavior. If behavior then shows a significant relationship to these clusters, it is clearly represented as a prominent network motif.

For each assay, the Akaike information criterion (AIC) was calculated, which balances both the fit of the data (log-likelihood) and the model complexity, resulting in an optimal number of clusters that reasonably fit the data but are not too numerous. To measure AIC for *k*-means, we assumed the clusters were Gaussian-distributed with identity covariance matrices. The formula was as follows:

$$\mathrm{AIC} = \mathrm{RSS}_{\min}(K) + 2MK$$

where RSS is the residual sum of squares, $M$ is the dimensionality of the dataset, and $K$ is the number of clusters.

For each implementation of *k*-means, clusters were identified using 10 different randomized initializations; the set with the minimum sum of Euclidean distances was used. The approach and avoidance clusters then identified, for each session, as those with, respectively, the highest and lowest mean avoidance/approach scores. The overall mean avoidance/approach scores for approach and avoidance clusters were then calculated across mice. To determine if these approach and avoidance cluster scores were statistically significant, the actual mean was compared to a sampling distribution

of means, calculated in an identical manner with permuted behavioral labels over 100 iterations. If the approach and avoidance score means were respectively greater than or less than 95% of this sampling distribution, they were considered significant. For the training/testing analysis, *k*-means was implemented on one assay as described above (the training assay), using only cells that co-registered between both assays. The cluster centroids identified in the training assay were then used to categorize approach and avoidance samples in the withheld testing assay. The mean avoidance/approach score was calculated for all approach and avoidance timepoints, across all training or testing sessions.

In a similar way, avoidance/approach states were identified by Hidden Markov Models (HMMs (4 states), using the top principal components of the neural data as input (accounting for $\geq 60\%$ of the total variance). Similar to *k*-means, this approach finds 'hidden states' or states represented by the neural data in an unsupervised manner; unlike *k*-means, HMMs additionally consider the sequentiality of the neural data. The parameters of the HMM were found via expectation maximization. These states were analyzed in an identical manner to the *k*-means clusters described above. For the code, see 'Expectation-Maximization for Hidden Markov Models using real-values Gaussian observations' at Zoubin Ghahramani's code base: http://mlg.eng.cam.ac.uk/zoubin/software.html.

## Statistical analysis

Significance values are included in the figure legends. Unless otherwise noted, all statistical comparisons were performed by either nonparametric Wilcoxon rank sum or signed-rank tests. With the exception of CoCA significance testing, all significance tests were two-tailed. Standard error of the mean was plotted in each figure as an estimate of variation of the mean. Correlations were calculated using Pearson's method. Multiple comparisons were corrected with the false discovery rate method. All statistical analyses were performed using SciPy (*Virtanen et al., 2020*) and custom MATLAB scripts.

## Acknowledgements

This work was supported by the National Institute for Mental Health (R00 MH106649 and R01 MH119089) (AA), the Brain and Behavior Research Foundation (Grants #22663, 27654, and 29204 respectively to AA, FMCVR, and JCK), the National Science Foundation (NSF-GRFP DGE-1650604, PJS), the UCLA Affiliates fellowship (PJS), and the Hellman Foundation (AA). Fundação de Amparo à Pesquisa do Estado de São Paulo (FAPESP), Research Grant #2019/17677–0 (JMI), #2014/05432–9 (NSC). FMCVR was supported with FAPESP Grants #2015/23092–3 and #2017/08668–1. MQLV was supported by the Achievement Rewards for College Scientists Foundation and by the National Institute for Mental Health award F31 MH121050-01A1.

## Additional information

### Funding

| Funder | Grant reference number | Author |
|---|---|---|
| National Institutes of Health | R00 MH106649 | Avishek Adhikari |
| National Institutes of Health | R01 MH119089 | Avishek Adhikari |
| Brain and Behavior Research Foundation | 22663 | Avishek Adhikari Fernando MCV Reis Jonathan C Kao |
| Brain and Behavior Research Foundation | 27654 | Fernando MCV Reis Avishek Adhikari Jonathan C Kao |
| National Science Foundation | DGE-1650604 | Peter J Schuette |
| Fundação de Amparo à Pesquisa do Estado de São Paulo | #2019/17677–0 | Juliane M Ikebara |
| Fundação de Amparo à Pesquisa do Estado de São Paulo | 2017/08668-1 | Fernando MCV Reis |

| | | |
|---|---|---|
| Fundação de Amparo à Pesquisa do Estado de São Paulo | 2014/05432-9 | Newton S Canteras |
| Hellman Foundation | | Avishek Adhikari |
| National Institutes of Health | F31 MH121050-01A1 | Mimi Q La-Vu |
| Achievement Rewards for College Scientists Foundation | | Mimi Q La-Vu |
| Brain and Behavior Research Foundation | 29204 | Avishek Adhikari<br>Fernando MCV Reis<br>Jonathan C Kao |

The funders had no role in study design, data collection and interpretation, or the decision to submit the work for publication.

### Author contributions
Fernando MCV Reis, Conceptualization, Supervision, Funding acquisition, Investigation, Writing - original draft, Writing - review and editing; Johannes Y Lee, Conceptualization, Formal analysis, Writing - original draft, Writing - review and editing; Sandra Maesta-Pereira, Formal analysis, Investigation, Writing - review and editing; Peter J Schuette, Formal analysis, Writing - review and editing; Meghmik Chakerian, Mimi Q La-Vu, Brooke C Tobias, Juliane M Ikebara, Investigation; Jinhan Liu, Formal analysis, Writing - original draft, Writing - review and editing; Alexandre Hiroaki Kihara, Funding acquisition, Methodology; Newton S Canteras, Conceptualization, Funding acquisition; Jonathan C Kao, Conceptualization, Resources, Formal analysis, Supervision, Funding acquisition, Methodology, Writing - original draft, Writing - review and editing; Avishek Adhikari, Conceptualization, Resources, Supervision, Funding acquisition, Investigation, Methodology, Writing - original draft, Project administration, Writing - review and editing

### Author ORCIDs
Fernando MCV Reis ⓘ https://orcid.org/0000-0002-0121-2887
Johannes Y Lee ⓘ http://orcid.org/0000-0003-2420-4916
Sandra Maesta-Pereira ⓘ http://orcid.org/0000-0001-6522-8311
Brooke C Tobias ⓘ http://orcid.org/0000-0003-2043-9523
Newton S Canteras ⓘ http://orcid.org/0000-0002-7205-5372
Jonathan C Kao ⓘ https://orcid.org/0000-0002-9298-0143
Avishek Adhikari ⓘ https://orcid.org/0000-0002-9187-9211

### Ethics
Animal experimentation: All procedures have been approved by the University of California, Los Angeles Institutional Animal Care and Use Committee, protocols 2017-011 and 2017-075.

### Decision letter and Author response
Decision letter https://doi.org/10.7554/eLife.64934.sa1
Author response https://doi.org/10.7554/eLife.64934.sa2

## Additional files
### Supplementary files
• Transparent reporting form

### Data availability
All data was uploaded to dryad and all code was uploaded to github. https://datadryad.org/stash/share/4GezSjw4dvDJClAWa_zRoNWioH9qzGtDCJjLQ89HVoA, https://doi.org/10.5068/D1TM2G, https://github.com/schuettepeter/eLife_dPAG-ensembles-represent-approach-and-avoidance-states (copy archived at https://archive.softwareheritage.org/swh:1:rev:7c5aa29ff557bd8955eafb0a5699960a79aee4f7).

The following dataset was generated:

| Author(s) | Year | Dataset title | Dataset URL | Database and Identifier |
|---|---|---|---|---|
| Reis FM, Lee JY, Maesta-Pereira S, Schuette PJ, Chakerian M, Liu J, La-Vu MQ, Tobias BC, Canteras NS, Kao JC, Adhikari A | 2021 | Dorsal Periaqueductal gray ensembles represent approach and avoidance states | https://datadryad.org/stash/share/4GezSjw4dvDJClAWa_zRoN-WioH9qzGtDCJjLQ89H-VoA | Dryad Digital Repository, 10.5068/D1TM2G |

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
