## [Decision Letter]

**Acceptance summary:**

The manuscript is a thorough examination of dPAG activity in threat exposure. Some strengths of the paper include relating the open-closed sensory states to threatening states in a live rat study, and the elegant neural analyses that show support for the generalization of neural representation of threat approach and avoidance across procedures.

**Decision letter after peer review:**

Thank you for submitting your article "Dorsal Periaqueductal gray ensembles represent approach and avoidance states" for consideration by *eLife*. Your article has been reviewed by 3 peer reviewers, including Mihaela D Iordanova as the Reviewing Editor and Reviewer #1, and the evaluation has been overseen by Laura Colgin as the Senior Editor. The following individuals involved in review of your submission have agreed to reveal their identity: Philip Jean-Richard-dit-Bressel (Reviewer #2); Jonathan Fadok (Reviewer #3).

The reviewers have discussed the reviews with one another and the Reviewing Editor has drafted this decision to help you prepare a revised submission.

Summary:

The manuscript looks at how dPAG neurons represent threat approach and avoidance using the elevated plus maze and a live rat exposure situation. The data show that the dPAG populations split in terms of avoidance and approach when the mice are exposed to the EPM. These populations correspond to threat approach and avoidance in the live rat procedure. Machine learning algorithms further show support for the generalization of neural representation of threat approach and avoidance across procedures. The paper uses established behavioral methods as well as state-of-the-art neural methods and analyses. It presents a thorough examination of dPAG activity in threat exposure tasks. Although the study of the dPAG in these tasks has been reported in the past (e.g., Deng et al., 2016, DOI: 10.1523/JNEUROSCI.4425-15.2016; Masberrer et al., 2020, DOI: https://doi.org/10.1523/JNEUROSCI.0761-18.2020), which limits the novelty of the present paper, relating the open-closed sensory states to threatening states in the live rat study is a nice development.

Essential revisions:

The existing data mostly supports the main conclusions of the paper and I do not think added experimentation is needed. However, there are some concerns:

1. Concerns about the analyses, interpretation and representation of the data:

a. An argument is made in terms of threat and operationalized as a threat score and dPAG activity. However, this threat score seems to be determined where in the EPM the mouse is and has nothing to do with perception of threat derived from a behavioural response. So, if differential dPAG neurons fire to open vs closed arms in the EPM, then it seems hardly surprising that there would be a correlations between the 'threat score' and neural activity. What is more the threat score is more related to distance from centre as opposed to any actual threat. The latter is inferred. Correlation with head dip may be a better examination.

b. Cells were defined by their bias in firing patterns in open vs closed arms, which was then used to predict animal location. It would be more compelling if cell classification was determined using separate portions of the session than that used for prediction (akin to way training and testing is separated for SVM) to avoid circularity.

c. Emphasis is placed on how activity patterns are shared between incompatible behaviours such as freezing and escape, both of which are reactions to threat. While this might suggest threat-coding, it may also be the case that shared activity are an artifact of the slow kinetics of GCaMP6s coupled with these behaviours occurring in close temporal proximity to each other. Is there any evidence that the same patterns of activity are observed when the potential contribution of other behaviours are factored out? Perhaps GLM could be used to isolate event-related activity kernels.

d. For Figure 2D, define what is meant by strong correlation. What is the threshold?

e. Please include the EPM aggregate activity heatmap for all cells, as is reported for the REA in Figure 3C.

f. Is the overall activity bias near the rat in Figure 3C driven by novel threat preference in previously 'neither' cells, reduced activation of closed cells near the safe wall, or increased activation of open cells near the rat (relative to activity during EPM)?

g. The data in Figure 3-S3A do not appear to match the data Figures 3C and 3F and therefore are not representative.

h. The increased z-score to rat movement by closed-arm cells in Figure 3I-J is inconsistent with the interpretation that these neurons reflect threat proximity, since the threat is moving closer to the mouse and these cells are inhibited in proximity to threat in other scenarios. The data are better explained by the activity representing the mouse's state on the exploratory-defensive continuum.

i. There should be a zoomed-out image of PAG histology showing the spread of the GCaMP expression and the lens placement.

2. Methods. Methods should be written rigorously enough to promote reproducibility. Some specific questions are included below, but the authors should ensure that they provide a thorough account of the methods used.

a. Lack of clarity regarding shuffling and bootstrapping. It is unclear whether the authors are in fact applying bootstrapping (resampling from dataset with replacement) or another analysis method such as permutation tests (randomly relabelling cases). Please clarify and provide more details within Methods.

b. k = 10 seems arbitrary. It would be useful to know the relative strength of k = 2-10+ clusters (e.g. table of mean and minimum silhouette values) to confirm k = 10 is sensible.

c. The behavioural timeline, methods and procedures were confusing. More detail and clarity need to be provided in the Methods. This should account for the differential number of animals in the EPM and REA. Further, it was unclear when habituation took place.

d. Is the threat score binary (+1 or -1) or is it bounded by -1 to +1, with values in between? If that latter, how are values in between calculated? A liner a relationship is assumed between threat and distance travelled, which seems appropriate for the open arm but perhaps more uniformly low throughout the closed arms.

3. Discussion.

a. A discussion of the results and how they fit into the larger body of PAG literature is necessary.

b. The significance of the neither cells being activated in the REA.

c. Any insight the authors may have into the circuit or physiological differences of the two populations.

d. A discussion of the counterintuitive finding that a greater relative ratio of open:closed neurons correlates with less time in the open arm and threat approach.

[Editors' note: further revisions were suggested prior to acceptance, as described below.]

Thank you for submitting your article "Dorsal Periaqueductal gray ensembles represent approach and avoidance states" for consideration by *eLife*. Your article has been reviewed by 3 peer reviewers, including Mihaela D Iordanova as the Reviewing Editor and Reviewer #1, and the evaluation has been overseen by Laura Colgin as the Senior Editor. The following individuals involved in review of your submission have agreed to reveal their identity: Philip Jean-Richard-dit-Bressel (Reviewer #2); Jonathan Fadok (Reviewer #3).

The reviewers agreed that the paper was greatly improved with the inclusion of necessary additions to methods, improvements to language, and addressing concerns around circular analyses and overlapping behaviour-related activity. The change in the semantics from EPM threat score to EPM location index was welcomed, although whether there is a linear gradient of threat in EPM remains up for debate. Some questions remain over the concerns raised previously. These are specified below.

Essential revisions:

1. Justification regarding the choice of 10 clusters has not been adequately addressed. A good solution would be to show that mean and minimum silhouette values for k = 10 is positive and higher/comparable to the range of other k-means solutions. This can be presented in a supplementary table. This is important because it goes a long way towards showing that the choice in clusters was valid. Currently the justification in the paper is not adequate and does not address why k=10 was chosen, rather is focused on why k=2 was not a good choice.

2. The added point that rat movements did not generally decrease distance to the mouse needs clarification. Did the increased distance between rat and mouse include movement made by the mouse? Given rat movement precipitates mouse escape, the increased distance is potentially due to mouse escape, not the rat moving away? Separately, was there a reason a 2 sec window was chosen? Rat movements towards the mouse followed by retreat (e.g. due to constrained movement) would easily elapse within 2secs, and be registered as movements away despite the mouse likely perceiving/responding to the advance. All relevant details about measuring rat movement should be included in Methods.

3. For variance thresholding, if this was performed per experimental session, were there cells that had high variance in one assay but low variance (and were thus excluded) in another assay? This would inflate the apparent degree of cross-assay coding. Details, i.e., a statement or Venn diagram, explaining how many cells were excluded on basis of EPM alone, REA alone, or both would be pertinent. If the number excluded on basis of only one assay is substantial, the text should acknowledge the impact this likely has on apparent cross-assay coherence.

---

## [Author Response]

Essential revisions:The existing data mostly supports the main conclusions of the paper and I do not think added experimentation is needed. However, there are some concerns:1. Concerns about the analyses, interpretation and representation of the data:a. An argument is made in terms of threat and operationalized as a threat score and dPAG activity. However, this threat score seems to be determined where in the EPM the mouse is and has nothing to do with perception of threat derived from a behavioural response. So, if differential dPAG neurons fire to open vs closed arms in the EPM, then it seems hardly surprising that there would be a correlations between the 'threat score' and neural activity. What is more the threat score is more related to distance from centre as opposed to any actual threat. The latter is inferred. Correlation with head dip may be a better examination.

The data in Figure 2E show that dPAG neural activity can be used to predict if the mouse is in the closed or open arms. Here, location is defined as a binary value, as the maze was divided into two locations (open and closed arms).

Our next analysis, in Figure 2F-G, shows dPAG activity can be used to predict not just which arm type the mouse is in, but where within that arm the mouse is. Is the mouse located in the beginning of the open arm (near the center) or is it in the end of the open arm? For this analysis the location of the mouse was not defined by a binary value (closed vs open), but with the Threat score, which is a continuous variable, corresponding to -1 in the end of the closed arms, 0 in the center and +1 in the end of the open arms. We created the EPM threat score to correlate neural activity with a continuous, non-binary, definition of mouse position. Figure 2F-G shows that dPAG neural activity does not simply encode arm type (as shown in 2E), but also finer grained location within an arm.

We believe this suggestion arose due to the name “EPM threat score”. This name suggests that the metric is directly reflecting threat, when in reality we simply used it as a number to describe the position of the mouse within the maze. We apologize for this misunderstanding, and thus have changed the name of the metric to “EPM location index”. We have altered the manuscript to remove any text that could indicate to the reader that this score is directly measuring threat.

The reason we investigated if the animal represents continuously varying locations within an arm type is because we believe that different locations within the open arm are more or less aversive, with the end of the open arms being the most aversive location in the maze. Our justification for this view is explained below.

As the reviewers correctly point out, the EPM location index is fully determined by the animal’s position in the maze, and as such this metric does not directly represent threat perception or behavioral response. However, we disagree that the location index has ‘nothing to do with perception of threat’. Regions of the maze with higher location index are correlated with higher avoidance, which is a behavioral response dictated by threat perception. Even within the open arms, mice avoid the regions with larger positive scores (corresponding to the end of the open arms) more than the regions with small positive scores, which are located in the beginning of the open arms, near the center of the maze. This behavioral gradient of avoidance within the open arms strongly indicates that the animal likely perceives higher threat in the extreme end of the open arms than in the beginning of the open arms. The most common interpretation of plus maze behavior is that mice avoid the open arms because they are more vulnerable to potential threats in open locations. This aversion facilitates thigmotaxis, a tendency to be close to vertical surfaces (i.e. closed arms). This view suggests that the beginning of the open arms are only slightly threatening, as if a dangerous stimulus is detected the mouse can quickly retreat to the safety of the closed arms. Conversely, it takes a longer time for the mouse to retreat to the closed arms if they are at the extreme end of the open arm. Thus, we argue that distance from the center of the maze is related to a gradient of threat, with the highest threat levels at the end of the open arms. The animal’s behavior supports this view, as mice avoid the extreme end of the open arm more robustly than the beginning of the open arm. We now include data showing that behavioral avoidance varies even within an arm type (Figure 2—figure supplement 1A, B), providing a stronger justification to investigate if dPAG cells represent finer-grained locations within an arm and discuss it in the text.

Although we support the interpretation that distance from center is a metric related to EPM aversiveness, we do not claim in the text that the EPM location index is directly reflecting threat perception or behavior, and we only used the EPM location index as a continuous, non-binary, metric to describe mouse location in the maze. Lastly, as mentioned above we changed the name of this score to “EPM location index” to avoid confusion.

It is unclear to us how to implement the Reviewer’s suggestion of using ‘head dips’ instead of the threat score. Head dips cannot be used as a gradual continuous metric of location within the maze.

b. Cells were defined by their bias in firing patterns in open vs closed arms, which was then used to predict animal location. It would be more compelling if cell classification was determined using separate portions of the session than that used for prediction (akin to way training and testing is separated for SVM) to avoid circularity.

We agree with this suggestion, and have therefore recalculated the results in Figure 2A-C, G to use the first 40% of each session for cell categorization and the last 60% for quantitative analyses. We altered the text and Methods to identify this. Interesting, we see similar results to before, indicating that cell responses to arm transitions and location in the EPM are robust during a session.

c. Emphasis is placed on how activity patterns are shared between incompatible behaviours such as freezing and escape, both of which are reactions to threat. While this might suggest threat-coding, it may also be the case that shared activity are an artifact of the slow kinetics of GCaMP6s coupled with these behaviours occurring in close temporal proximity to each other. Is there any evidence that the same patterns of activity are observed when the potential contribution of other behaviours are factored out? Perhaps GLM could be used to isolate event-related activity kernels.

This is an important point that was not addressed in the original submission. We thank the Reviewer for this question. In the rat assay the mouse is initially placed far from the rat. The mouse spends most of the time far from the rat, but sometimes it slowly approaches the rat, and then escapes in high velocity back to the safety of the opposite side of the chamber. After reaching the safer side, the mouse often displays many freezing bouts. Thus, freezing tends to occur after escaping. The analysis showed that the transition probability from freezing to escape is much lower than escape to freezing (0.01 and 0.61 respectively, n=44 mice). Thus, freezing tends to happen after escape (but escape rarely happens immediately after freezing).

As a consequence, it is possible that the freeze-related activity we identify could reflect slowly decaying GCaMP activity from a prior recent escape, rather than new “de novo” activity induced by freezing. To account for this confound, we calculated freeze bout-triggered activity traces excluding all freezes that happen within 10 seconds from an escape. These data show that even when excluding potentially confounding escape-related activity prior to freezing, closed cells and open cells show respective increases and decreases in activity during freezing (new Figure 3, panels D and E; I and J).

d. For Figure 2D, define what is meant by strong correlation. What is the threshold?

We apologize for not defining ‘strong’. We did not wish to create a binary distinction between cells with strong and weak correlation. Rather, we intended to emphasize that speed cannot explain the arm-type preference displayed by dPAG cells. We altered the text to read “Importantly, 95% of dPAG cells showed relatively low correlations of speed and neural activity, between -0.17 and +0.2. These results suggest that arm-related activity preferences are not driven by variations in velocity”.

e. Please include the EPM aggregate activity heatmap for all cells, as is reported for the REA in Figure 3C.

We have performed this new analysis as requested and added it to the manuscript (Figure 2—figure supplement 1). Aggregate dPAG does not show differential activation in closed and Open arms, as shown in Figure 2—figure supplement 1C.

f. Is the overall activity bias near the rat in Figure 3C driven by novel threat preference in previously 'neither' cells, reduced activation of closed cells near the safe wall, or increased activation of open cells near the rat (relative to activity during EPM)?

We thank the Reviewer for the opportunity to address this important question. We show in Figure 3F that all three cell types are more active near the rat than far from it. However, consistent with their activity patterns in the EPM, open cells show significantly higher activation near the rat than closed cells.

g. The data in Figure 3-S3A do not appear to match the data Figures 3C and 3F and therefore are not representative.

We apologize for this oversight. We now updated Figure 3-S3A to show representative examples of open and closed cells respectively displaying increases and decreases in activity as the mouse approaches the rat. This same result is shown averaged across cell ensembles in Figure 3G.

Figure 3C shows that dPAG cells in general (without separating into closed and open cells) are more active near the rat. This result is independent of the example individual heat maps for closed and open cells in Figure 3-S3A, and thus is not in disagreement (or agreement) with 3-S3A. Different results can be found when separating the cells into closed and open ensembles compared to pooling all cells together.

If the explanation above is not satisfactory, we ask the Reviewer to please explain in more detail what the discrepancy is between 3-S3A and 3C.

h. The increased z-score to rat movement by closed-arm cells in Figure 3I-J is inconsistent with the interpretation that these neurons reflect threat proximity, since the threat is moving closer to the mouse and these cells are inhibited in proximity to threat in other scenarios. The data are better explained by the activity representing the mouse's state on the exploratory-defensive continuum.

We apologize for not explaining more clearly the rat movement metric. As stated in the paper, the rat is constrained in its movement by a harness, which only allows it to move in a restricted area. Thus, for the most part, the rat cannot significantly change the distance to the mouse by moving. We now added a new panel in Figure 3—figure supplement 4 (panel B) showing that proximity to the rat does not generally increase during rat movements. The rat movement bouts are mostly changes from one resting position to another, or exploratory sniffing towards the walls, and may be in any direction. The rats were screened for low aggressiveness prior to their use, and consequently rat movement is very rarely motion towards the mouse, and the rat never displayed any attacking bouts.

Even though rat motions are not directed towards the mouse, the mouse tends to react to rat movements as sign of increased threat, and thus rat movement predicts decreased approach to rat, and increased escape and freezing (Figure 3-S4). These data show that rat movement is predictive of increased defensive behaviors such as escape and freezing. We argue that closed cells are more activated during threat-avoidance related defensive behaviors (i.e., escape and freezing). Consequently, it is expected that rat movement (which is a stimulus that provokes more escape and freezing) should be positively correlated with increases in closed cell activity.

Nevertheless, we strongly agree with the Reviewer’s view that the data is well explained by changes in the “exploratory-defensive continuum”. This continuum is reflected in both behavioral and non-behavioral features. Higher defensive states are displayed during increased threat perception, and one of the metrics influencing threat perception is threat distance, while another is movement of the predator.

i. There should be a zoomed-out image of PAG histology showing the spread of the GCaMP expression and the lens placement.

We agree with this comment and have added a zoomed out histology photo showing GCaMP expression spread and lens placement (Figure 1—figure supplement 1).

2. Methods. Methods should be written rigorously enough to promote reproducibility. Some specific questions are included below, but the authors should ensure that they provide a thorough account of the methods used.

We have significantly increased the level of detail in the methods to ensure reproducibility. In addition, all scripts have been posted on github.

a. Lack of clarity regarding shuffling and bootstrapping. It is unclear whether the authors are in fact applying bootstrapping (resampling from dataset with replacement) or another analysis method such as permutation tests (randomly relabelling cases). Please clarify and provide more details within Methods.

We apologize for the lack of clarity. The analyses were incorrectly described as using bootstrap resampling when they were permutation tests using randomly permuted labels. We have corrected the text and Methods to reflect this.

b. k = 10 seems arbitrary. It would be useful to know the relative strength of k = 2-10+ clusters (e.g. table of mean and minimum silhouette values) to confirm k = 10 is sensible.

We repeated the analysis for a range of *k* values and now summarize these results in the revised Figure 4—figure supplement 1. The mean avoidance and approach score values were respectively and significantly less than and greater than the distribution of avoidance/approach values built from permuted neural data for EPM and rat assay across a range of *k* values (EPM: *k*=2-12; rat: *k*=3-12). While *k*-means with *k*=2 did not significantly differentiate the avoidance score from the permuted distribution in the rat assay, this is not particularly surprising--the analysis is unsupervised and the most salient dPAG feature may not be avoidance/approach for this assay. However, given an additional cluster (*k*=3), the avoidance score fell significantly below the permuted distribution for the rat assay. The approach and avoidance clusters defined on one assay were, moreover, significantly different when applied to the other assay across this range of *k* values. This analysis worked for all *k* values, from 3 to 12. To avoid plotting so dozens of plots demonstrating the validity across k values, we show these results only for the minimum and maximum significant outputs of the tested range (Author response image 1).

**Author response image 1. sa2fig1:** Approach and avoidance identified by k-means with a range of *k* values and Hidden Markov Model. (A) (top) Arrow depicts the mean avoidance/approach score for avoidance and approach clusters (3 clusters), identified by *k*-means, across concatenated sessions (*n* = 7 mice). This mean was compared to a permuted distribution of avoidance/approach means, calculated by shuffling the neural data 100 times (n cells same as Figure 4C; *p* < 0.01). (bottom left) Bars depict the mean Rat and EPM avoidance/approach scores (+/- 1 s.e.m.) for approach and avoidance clusters across mice. (bottom right) As described in Methods and Figure 4B, these cluster centroid locations, trained on one assay, were then used to define approach and avoidance timepoints in the other assay. Bars depict the corresponding mean avoidance/approach score (+/- 1 s.e.m.) for this testing data. (Train on EPM: avoidance cluster *n* = 17753, approach cluster n= 13096; Test on Rat: avoidance cluster *n* = 15727, approach cluster *n* = 7724; Train on Rat: avoidance cluster *n* = 21092, approach cluster *n* = 10672; Test on EPM: avoidance cluster *n* = 9867, approach cluster *n* = 3028; n cells same as Figure 4D; Wilcoxon ranked-sum test) (B) Same as (A), but with 12 clusters. (Train on EPM: avoidance cluster *n* = 2614, approach cluster n= 3671; Test on Rat: avoidance cluster *n* = 1749, approach cluster *n* = 1488; Train on Rat: avoidance cluster *n* = 3693, approach cluster *n* = 2337; Test on EPM: avoidance cluster *n* = 2163, approach cluster *n* = 1323; n cells same as Figure 4D; Wilcoxon ranked-sum test) *** *p* < 0.001.

c. The behavioural timeline, methods and procedures were confusing. More detail and clarity need to be provided in the Methods. This should account for the differential number of animals in the EPM and REA. Further, it was unclear when habituation took place.

We agreed with the Reviewer and rewrote the behavioral timeline and methods in the Methods session in this new version of the manuscript.

d. Is the threat score binary (+1 or -1) or is it bounded by -1 to +1, with values in between? If that latter, how are values in between calculated? A liner a relationship is assumed between threat and distance travelled, which seems appropriate for the open arm but perhaps more uniformly low throughout the closed arms.

The threat score (now called EPM location index) is bounded between -1 to +1. A value of zero is assigned to the center of the maze, and values were linearly interpolated from zero to -1 or +1 respectively in the end of the closed arms and open arms. We are not assuming a linear distance between threat and distance, even though we agree that higher score values are associated with higher threat. This score was developed only to have a continuous metric of plus maze location to test if dPAG neural activity can predict location within an arm type, as explained in question “1A” above. We altered the text to make it clear that this score is only being used as a continuous metric of location within the maze, and we are in no way implying this score is a direct measure of threat perception. Please see our detailed explanation in question 1a above,

3. Discussion.a. A discussion of the results and how they fit into the larger body of PAG literature is necessary.

As pointed out by the Reviewers earlier, prior papers have recorded dPAG activity during threat exposure. We now discuss this prior work in more detail and explain how our data advance understanding of the dPAG. We believe the novelty of this paper is in:

1. Tracking the same cells across two assays and finding that dPAG represent open spaces and proximity to a predator with shared activity patterns

2. Performing at population-level analysis of dPAG cells, rather than only quantifying the % of cells that are activated or inhibited by specific behaviors

3. Demonstrating that dPAG cells are not only encoding specific behaviors, but rather that dPAG ensemble activity reflects moment-to-moment changes in the approach-avoidance state of the animal. This is our conceptually most important result, and it indicates that the view that the dPAG is merely a motor output structure is likely oversimplified.

We have now altered the manuscript, discussing these findings in the context of prior papers.

b. The significance of the neither cells being activated in the REA.

Our interpretation of these data is that the neither ensemble may require a higher amount of threat to be activated. The relatively low threat open arms are not enough to activate them, while proximity to a live predator is more dangerous, and thus is enough to activate these cells. We have added this discussion to the text.

c. Any insight the authors may have into the circuit or physiological differences of the two populations.

We speculate that perhaps these two cell classes may correspond to populations expressing different dPAG markers, such as substance P, nitric oxide or cholecystokinin. Another possibility is that these two cell classes have distinct connectivity patterns. We added discussion related to this issue in the text, while making it clear that this work does not answer this question, and future studies are needed to further dissect dPAG cell types based on genetic and connectivity differences.

d. A discussion of the counterintuitive finding that a greater relative ratio of open:closed neurons correlates with less time in the open arm and threat approach.

Our results are very similar to the one previously reported by Jimenez et al., 2018 in Neuron (“Anxiety Cells in a Hippocampal-Hypothalamic Circuit”). In that paper the authors identify ventral hippocampal cells that are preferentially active in the open arms of the elevated plus maze. Increased open arm activity was correlated with decreased open arm exploration and inhibition of the ventral hippocampus (which would result in inhibition of these open arm cells) increased open arm exploration. Thus, both this published result in the hippocampus as well as our dPAG result suggest that increased activity of open arm cells is correlated with decreased open arm exploration.

In our view, a possible interpretation for these results is that rather than causing open arm exploration, open arm ensemble is highly responsive to risk-assessment and proximity to threat. Considering this hypothesis, the present results suggest that mice with more open arm cells display an increased sensitivity to risk-evaluation reflected on increased avoidance of threat. We now added text discussing our interpretation of this result.

[Editors' note: further revisions were suggested prior to acceptance, as described below.]

Essential revisions:1. Justification regarding the choice of 10 clusters has not been adequately addressed. A good solution would be to show that mean and minimum silhouette values for k = 10 is positive and higher/comparable to the range of other k-means solutions. This can be presented in a supplementary table. This is important because it goes a long way towards showing that the choice in clusters was valid. Currently the justification in the paper is not adequate and does not address why k=10 was chosen, rather is focused on why k=2 was not a good choice.

To address the Reviewers' concern regarding our choice of cluster number, we have added Figure 4—figure supplement 1, which depicts the mean (+/- s.e.m.) Akaike information criterion (AIC) for 2 to 10 clusters. The AIC score is a standard hyperparameter and model selection metric in machine learning. We chose the AIC compared to silhouette values because the AIC score prioritizes model fit and complexity, rather than cluster consistency. The AIC measures the log likelihood minus a model complexity term that reflects the number of model parameters. The AIC therefore balances both the fit of the data (log-likelihood) and the model complexity, resulting in an optimal number of clusters that fit the data reasonably but are not too numerous. To measure AIC for k-means, we assumed the clusters were Gaussian distributed with identity covariance matrices.

This supplemental figure shows that the data is best fit by a k-means model with 2 clusters for the EPM assay and either 2 or 3 clusters for the rat exposure assay, due to their significantly lower AIC than other tested values of *k*.

We thus no longer use 10 clusters, and instead updated Figure 4C-D and Figure 4-2 with the optimized *k* values mentioned above.

2. The added point that rat movements did not generally decrease distance to the mouse needs clarification. Did the increased distance between rat and mouse include movement made by the mouse? Given rat movement precipitates mouse escape, the increased distance is potentially due to mouse escape, not the rat moving away? Separately, was there a reason a 2 sec window was chosen? Rat movements towards the mouse followed by retreat (e.g. due to constrained movement) would easily elapse within 2secs, and be registered as movements away despite the mouse likely perceiving/responding to the advance. All relevant details about measuring rat movement should be included in Methods.

The distance presented did not include movement by the mouse and was relative to the mouse's position at the start of movement. Consequently events categorized as increasing rat to mouse distance are due to the rat moving away from the mouse, rather than due to the mouse escaping.

We added a polar plot displaying the angle of all rat movements relative to the mouse (Author response image 2 and also Figure 3, figure supplement 4C). An angle of zero corresponds to Rat movements towards the mouse, while an angle of 180 indicates movements away from the mouse. Note that movements away from the mouse are more common than movements towards the mouse. Taken together, these data show that the rat is not consistently moving towards the mouse, as indicated both by the polar angle plot and also by the distribution of mouse-rat distance changes during rat movements (Figure 3, figure supplement 4B).

A window of 2 seconds was used to allow sufficient time to include the entire rat movement bout. We also added detailed information in the Methods explaining the measurements of rat movements.

**Author response image 2. sa2fig2:** Rat movements are not consistently directed towards mice. (A) Polar plot showing the distribution of direction of rat movement relative to mouse position at time = 0 seconds. (B) Distribution of direction of average rat movement across 2 seconds after rat movement initiation relative to mouse position at the start of rat movement initiation. Zero degrees represents movement toward initial mouse position, while 180 degrees corresponds to movements away from the mouse.

3. For variance thresholding, if this was performed per experimental session, were there cells that had high variance in one assay but low variance (and were thus excluded) in another assay? This would inflate the apparent degree of cross-assay coding. Details, i.e., a statement or Venn diagram, explaining how many cells were excluded on basis of EPM alone, REA alone, or both would be pertinent. If the number excluded on basis of only one assay is substantial, the text should acknowledge the impact this likely has on apparent cross-assay coherence.

Among the 375 total co-registered cells, 10 were excluded because of low variance in both the EPM and Rat, 11 were excluded because of low variance in the EPM but were not excluded in the Rat, and 38 were excluded in the Rat but not the EPM. We opted to show a more informative color scatterplot instead of the suggested Venn diagram because the full range of the data can be seen in the scatterplot. This scatterplot (Author response image 3) reveals that even among cells that were excluded due to low variance in only one assay, very few cells had very large variance in the other assay. For example, very few of the green cells (excluded due to low variance in the EPM) had very high variance in the Rat assay.

Importantly, our metric of cross-assay representation is based on the same cells similarly representing concrete threat-related metrics across assays, such as avoidance or approach behaviors. Our metric of cross-assay representation is not a direct consequence of simply having overall higher or lower variance across assays. Thus, it is unlikely that the selection of cells based on variance could be artifactually increasing cross-assay representation.

Instead of mentioning in the text that the selection of cells could be influencing cross-assay coherence, as suggested by the Reviewers, we opted instead to directly demonstrate that all of our main results can be replicated without excluding any cells. These data show that cell selection by variance thresholding does not artifactually create our results.

**Author response image 3. sa2fig3:** Raw variances of co-registered cells for both EPM and Rat assays across each entire session. (n=375 cells).